# Analyzing information sharing behaviors during stance formation on COVID-19 vaccination among Japanese Twitter users

**Sho Cho**[1][☯]*, **Shohei Hisamitsu**[1][☯][¤], **Hongshan Jin**[2], **Masashi Toyoda**[2], **Naoki Yoshinaga**[2]

**1** The University of Tokyo, Bunkyo-ku, Tokyo, Japan, **2** Institute of Industrial Science, The University of Tokyo, Bunkyo-ku, Tokyo, Japan

☯ These authors contributed equally to this work.
¤ Current address: Microsoft, Minato City, Tokyo, Japan
* cs@tkl.iis.u-tokyo.ac.jp

**Data Availability Statement:** The Twitter data cannot be shared publicly due to Twitter's (X Corp.) restrictions on redistributing Twitter Content to third parties (https://developer.twitter.com/en/

## Abstract

To prevent widespread epidemics such as influenza or measles, it is crucial to reach a broad acceptance of vaccinations while addressing vaccine hesitancy and refusal. To gain a deeper understanding of Japan's sharp increase in COVID-19 vaccination coverage, we performed an analysis on the posts of Twitter users to investigate the formation of users' stances toward COVID-19 vaccines and information-sharing actions through the formation. We constructed a dataset of all Japanese posts mentioning vaccines for five months since the beginning of the vaccination campaign in Japan and carried out a stance detection task for all the users who wrote the posts by training an original deep neural network. Investigating the users' stance formations using this large dataset, it became clear that some neutral users became pro-vaccine, while almost no neutral users became anti-vaccine in Japan. Our examination of their information-sharing activities during a period prior to and subsequent to their stance formation clarified that users with certain types and specific types of websites were referred to. We hope that our results contribute to the increase in coverage of 2nd and further doses and following vaccinations in the future.

## Introduction

Vaccination is thought to be one of the most powerful measures for containing outbreaks of infectious diseases such as COVID-19. When infectious diseases spread, wide acceptance of vaccines is necessary to prevent the spread of epidemic diseases. Since the 2019 novel coronavirus disease (hereinafter called COVID-19) pandemic, many countries recognized low vaccination coverage as a significant challenge [1].

In Japan, a country that had previously ranked among the nations with the lowest vaccine confidence globally [2], significant concerns existed regarding vaccine uptake, particularly among young individuals. According to a national survey on the intent to vaccinate against COVID-19 in February 2021, 32.9% answered "uncertain," and 11.0% answered "no" regarding vaccination. However, following the beginning of the vaccination campaign on February

developer-terms/agreement-and-policy). The tweet contents are accessible via the Twitter API provided by X Corp (https://developer.twitter.com/en). Additionally, our stance annotation data cannot be shared publicly. Instead, researchers may request our models solely for non-commercial purposes, specifically to reproduce our experimental results in their research. We provide two model data: a BERT-based language model which has undergone additional training on the Twitter data, and a stance classification model built upon the language model. These include codes and trained parameters. For any inquiries, please reach out to our group representative at contact@tkl.iis.u-tokyo.ac.jp.

**Funding:** This research was conducted as part of "COVID-19 AI & Simulation Project" run by Mitsubishi Research Institute commissioned by Cabinet Secretariat, JAPAN. The methods for analysis were developed with support from JST CREST Grant Number JPMJCR19A4 and JSPS KAKENHI Grant Number JP21H03445. The funders had no role in study design, data collection and analysis, decision to publish, or preparation of the manuscript.

**Competing interests:** The authors have declared that no competing interests exist.

17, 2021, the percentage of individuals who received a second dose of the vaccine (fully vaccinated at the time) in Japan saw a remarkable surge, rising from 3.4% in June to 75.3% in October of the same year. This rapid increase put Japan in first place among G7 nations (Japan, Canada, Italy, France, United Kingdom, Germany, and the United States), as depicted in Fig 1. The vaccination campaign in Japan thus proceeded smoothly, with many individuals who initially hesitated to get vaccinated consequently opting to receive the vaccine.

Examining successful vaccination campaigns, such as those in Japan, is crucial for understanding common factors influencing acceptance or hesitancy toward vaccines, and recently much research has studied these factors using social media data [4–14] while conventional approaches often rely on questionnaires and surveys [1, 2, 15, 16]. These investigations found several common reasons for vaccine hesitancy: anxieties about vaccine safety [4] and a lack of trust in vaccine efficacy [5]. Other studies looked into people's attitudes towards vaccination, exploring factors such as cross-country variations in positive and negative views on vaccination [9], spatiotemporal shifts in sentiment towards vaccines in the US [8], and the polarization among different vaccination communities in Japan [6]. A later study found that the majority of negative sentiment in Twitter predominantly mentioned the coercive policies or vaccine mandates, rather than safety or efficacy concerns [10]. Several studies reported that SNS users' stances toward vaccination were characterized by the information they obtained from SNS, such as external links [11, 12] or posts from other users [13, 14], resulting in wrong medical stances.

While these studies have enriched our understanding of overall characteristics and trends of public sentiment regarding COVID-19 vaccination, they assume that the stances of SNS users remain fixed throughout the study period, neglecting the process of stance formation of each user. We consider the stance formation as the process where users, initially without any stances on COVID-19 vaccination, shape their perspectives. In this study, users are classified into three stances: Pro-vaccine (those accepting or planning/getting vaccinated), Anti-vaccine (those avoiding vaccination), and Neutral (those without a specific stance on vaccination). We mainly focused on SNS users who were initially neutral and eventually held Pro-vaccine or Anti-vaccine stances and examined the contents shared by these users, including tweets from other users and external websites during their stance formation.

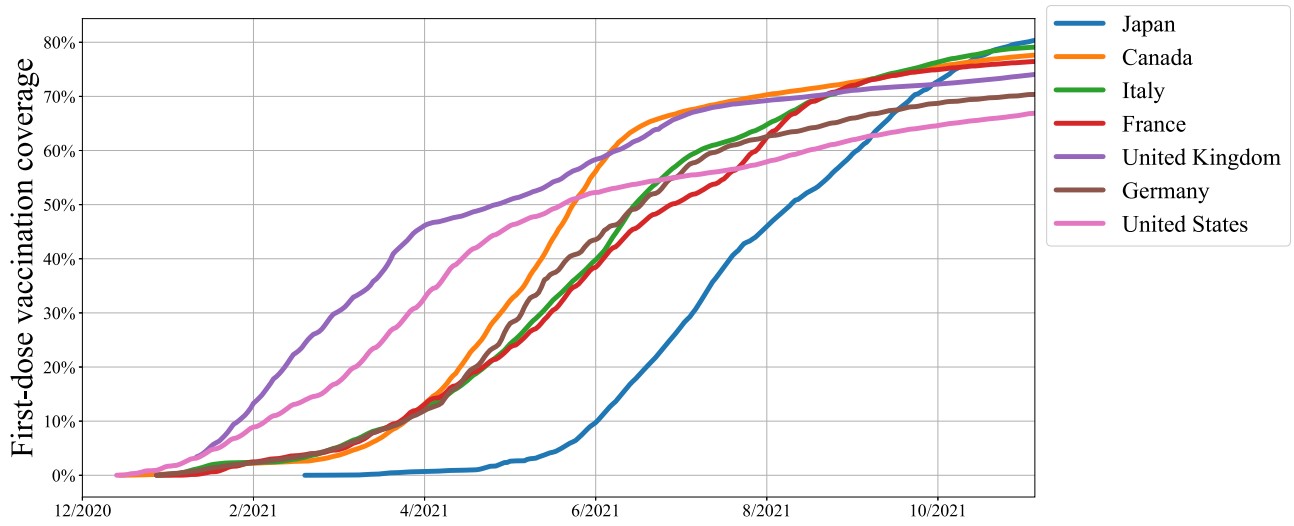

**Fig 1. Full vaccination coverage (1st & 2nd doses) of G7 countries based on data provided by Mathieu et al. [3].**

To draw lessons from Japan's success in improving COVID-19 vaccine coverage, we conducted an analysis of users' vaccination stances on Twitter for the purpose of identifying information-sharing behaviors that might have impacted their stance formations. We collected all Japanese tweets including a reference to vaccination over a span of five months when vaccine coverage rapidly grew (from June to October in 2021), which constituted a dataset of vaccination-related tweets. Within this dataset, we annotated a portion of the tweets with the stances of the users who posted them. Using this annotated dataset, we trained a classifier for stance detection, which is an NLP task [17] for predicting a stance (typically favor, against, and none) of given text toward a certain target. We developed a deep neural network utilizing both content and network features to assign vaccination stances to each tweet in the dataset and determined each Twitter user's stance toward COVID-19 vaccines. Using this classifier, we periodically aggregated the predicted stances for each user to investigate how users' stances on vaccination were shaped over time. This paper represents an extended version of our previously published work [18], in which we have enhanced the performance of our deep neural model and incorporated several new results into our analysis.

Our analysis is conducted as follows. We first examined the shift in stance distribution, discovering that the pro-vaccine group significantly outnumbered the anti-vaccine group. Subsequently, we investigated polarization in the user reaction graphs, and as a result, there was a gradual increase in the degree of polarization between the two factions over time. This finding suggested that the impact of anti-vaccine activity in Japan was limited. Finally, we focused on the users who were neutral at the beginning of the period, became pro-vaccine or anti-vaccine, and kept the stances. Our deeper exploration into their information-sharing behaviors allowed us to find potential factors that influenced their stance formation. One such user group, i.e., users who moved from neutral to pro-vaccine, exhibited a preference for more reliable information sources. These users frequently engaged with accounts of medical doctors and mainstream media outlets and shared links to external sites of mass media and web news. On the contrary, users who moved from neutral to anti-vaccine showed a tendency for alternative information sources. They regularly interacted with accounts of unclear or unknown occupation, and commonly shared links to bulletin board systems (BBSes), weblogs, and video hosting sites.

## Related work

Research into public attitudes towards vaccination, particularly in relation to COVID-19, has leveraged social media data [7]. Using data from social media has many merits compared with traditional survey-based methods. For example, they make it possible to timely observe public opinions, leading to better understanding of vaccination intentions and attitudes regarding ongoing immunization campaigns [19]. We divided vaccine-related research using social media data into three categories: thematic analyses of vaccine-related discussions, assessments of polarization in vaccine debates, and observations of stance formation in vaccination attitudes.

Thematic analyses have been used to identify prevalent topics concerning vaccines and to investigate the underlying causes of vaccination hesitancy. In such work, anxiety about vaccine safety [4] and doubt in vaccine efficacy [5] were commonly regarded as the most common reasons. Research focused on polarization within vaccine debates has quantified the ratio of positive to negative perspectives on vaccine-related content [9], examined the network structure [20], and assessed the degree of polarization among different vaccine communities on social networks [6]. In addition, some studies have analyzed sentiment changes in response to specific vaccine-related events to find the topics influencing vaccination intentions. Hu et al. [8]

conducted a study on the spatiotemporal patterns of public sentiment and emotion, tracking these factors over time at both national and state levels in the United States. They identified three distinct phases during the pandemic period where sharp changes in public sentiment and emotion occurred. It is noteworthy, however, that these studies on online vaccine debates [5, 20] did not focus the formations of users' stances on vaccinations.

A variety of studies incorporated various techniques such as sentiment analysis, stance detection, and graph analytics. The majority of these studies assigned sentiment labels to each tweet using classifiers that were trained on a small set of accurate labels while some studies preferred to use crowd-sourcing platforms like Amazon Web Services (AWS) for labeling tweets [21]. Yousefinaghani et al. [22] used the Valence Aware Dictionary and sEntiment Reasoner (VADER) [23], a lexicon and rule-based sentiment analysis tool written in Python. Cotfas et al. [24] used several machine learning and deep learning methods to classify users' stances towards vaccination. They demonstrated that the use of Bidirectional Encoder Representations from Transformers (BERT) [25] showed state-of-the-art results. Alhuzali et al. [26] combined sentiment analysis with geographical analysis. They used tweets from various cities in the United Kingdom to predict sentiment labels with a deep learning model and performed city-wise analyses. Mønsted et al. [14] leveraged transfer learning to enhance their stance classifier's performance. They studied the propagation process of misinformation about vaccination on a mutual mention/retweet network. Garcia et al. [27] compared the transition of COVID-19 related topics in Brazil with that of the USA in Twitter. They also combined deep learning models and embedding-based techniques. To measure the level of polarization, some techniques for graph analytics such as community detection like METIS [28] and stochastic simulation like Random Walk Controversy (RWC) [29] were used. For instance, Yuan et al. [20] monitored polarization in debates on vaccination using the Louvain method [30], while Miyazaki et al. [6] quantified the degree of polarization using RWC.

This paper explores the information-sharing behaviors of SNS users during their stance formations for a more profound understanding of such behaviors. In terms of methodology, our study differs from prior work, which typically relies solely on content for sentiment analysis and stance classification. Instead, we propose a novel deep learning model incorporating both content and network-based elements. Furthermore, we leverage established methods of social network analysis to quantify controversies surrounding vaccine debates. This allows us to observe the formation of stances within the context of polarization.

## Dataset construction

Twitter is a social media platform widely used in Japan and has a broad range of age groups, particularly younger generations (https://www.humblebunny.com/japans-top-social-media-networks). This platform allows users to share their thoughts through tweets and engage with others by reacting to their posts. These reactions, comprising retweets, quotes, and replies, indicate the users' interest in the content of the tweets. Consequently, we leveraged these tweets to monitor people's attitudes towards vaccination and used the reactions to look into their information-sharing behaviors.

Vaccination for individuals aged 18 to 64 in Japan started on June 17, 2021. By the end of October, approximately 75% of the population had been fully vaccinated. From all Japanese tweets, we extracted tweets from June 1, 2021 to October 31, 2021 that contained the keyword "ワクチン" (wakuchin, vaccine in English), resulting in 19,502,448 tweets posted by 4,446,499 users. The data of all Japanese tweets was provided by the NTT Data Japan Corporation. We took the utmost care in handling personal data and conducted our research in strict compliance with Twitter's "Twitter Developer Agreement"(https://developer.twitter.

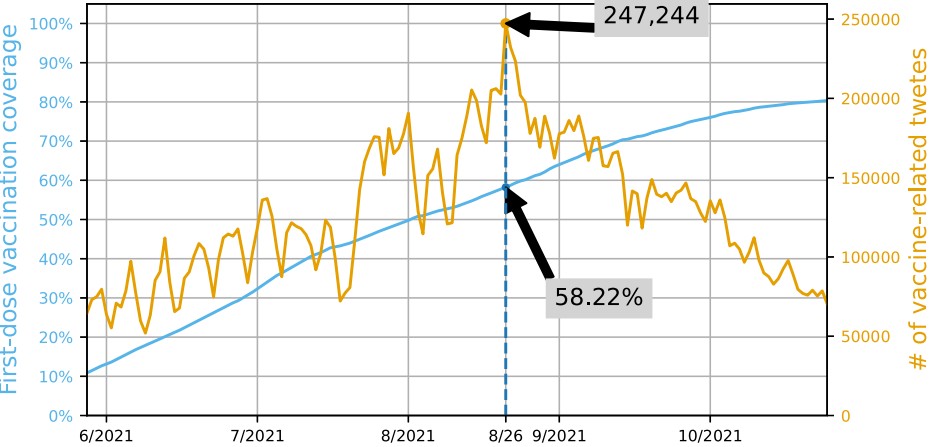

**Fig 2. Changes in first-dose vaccination coverage [3] and number of vaccine-related tweets in Japan.**

com/en/developer-terms/agreement-and-policy/source). We intentionally excluded tweets posted by the "share via Twitter" function on certain websites and those produced by an app named "shindanmaker," as they lacked any valuable information for assessing user intent. Our investigation of user accounts revealed that tweets posted from 4 major clients (iPhone, Android, iPad, and Web) accounted for 93.9%, and tweets posted from client applications including "bot" in their name accounted for only about 1%, which meant that explicit bot accounts had only a small impact on our analysis. Consequently, we created a dataset of vaccine-related tweets, consisting of 18,462,168 tweets posted by 4,408,669 unique users. Fig 2 illustrates the trend in the number of collected tweets. There was a great increase in tweet volume from June, which began to decline in late August when the initial vaccination rate surpassed 58.22%.

## Vaccination stance classification

As the basis and key to subsequent analysis, we identified users' stances towards vaccination. Stance detection was done by using a vaccination stance classifier trained on our annotation dataset. This dataset was constructed on vaccine-related tweets manually annotated by four annotators. We trained the classifier using this dataset to label the other tweets. Fig 3 shows our tweet-selection process.

Previous studies on predicting vaccine stances from tweets [20, 24] relied only on textual information and failed to classify tweets that referred to posts with the opposite vaccination stance. We additionally used reaction graph information to classify tweets into users' stances towards vaccination using a deep neural network.

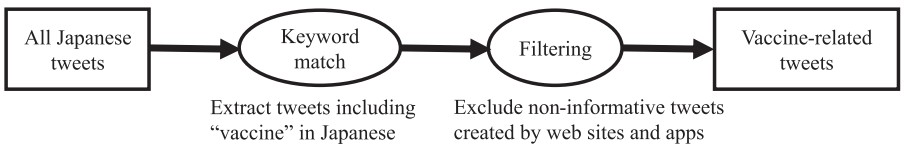

**Fig 3. Outline of our tweet-selection process.**

**Table 1. Annotation criteria and example tweets.** Example tweets are translated from Japanese.

| Labels | Criteria | Example tweets (translated) |
|---|---|---|
| Pro-vaccine | Saying the user was vaccinated | I got the first dose of the vaccine |
| | Saying the user planned to get vaccinated | I've finally reserved my COVID-19 Vaccinations. |
| | Recommending vaccination to others | Get vaccinated! |
| | Criticizing anti-vaccine people | Are anti-vacs all idiots? |
| Neutral | Showing facts (e.g., number of vaccinated) | 86% of people say their arm hurts after injection. |
| | Introducing press release from public institutions | According to the MHLW website, antibodies are completed in 1 week to 2 weeks. |
| | Discussing topics irrelevant to the pros and cons of vaccination | Everyone has got vaccinated? |
| Anti-vaccine | Expressing intention not to get vaccinated | I won't get vaccinated. |
| | Calling attention to not getting vaccinated | You shouldn't vaccinate. |
| | Criticizing pro-vaccine people | Idiots who wanted to be vaccinated were lining up at the hospital. |

## Annotation of stance of tweets

In this study, users are classified into three stances based on their stance towards vaccination: Pro-vaccine (those accepting or planning/getting vaccinated), Anti-vaccine (those avoiding vaccination), and Neutral (those without a specific stance on vaccination). To develop a deep learning model for stance classification, we conducted manual annotation on a subset of vaccine-related tweets. This involved categorizing tweets into three classes: pro-vaccine, anti-vaccine, and neutral towards vaccines. Criteria for stance annotation were established (see Table 1) to ensure consistent annotation. Pro-vaccine tweets consisted of expressions of support for vaccines, personal vaccination plans or experiences, recommendations for vaccination, and criticism of anti-vaccine sentiments. Anti-vaccine tweets consisted of vaccine denial, discouragement of vaccination, and criticism of pro-vaccine advocates. Neutral tweets consisted of factual information, introductions to press releases from public institutions, and discussions unrelated to the merits and drawbacks of vaccination. In accordance with these annotation criteria, four annotators labeled the same 500 tweets to measure the inter-annotator agreement; Fleiss' kappa coefficient [31] for this annotation task was 0.74, which confirms the stability of the annotations. Each annotator then labeled an average of 2313 randomly-chosen tweets, resulting in a total of 9250 labeled tweets. Fig 4 shows our annotation, learning, and classification process.

## Text and graph-based stance classification

With the above annotated tweets, we next trained a deep neural network on the basis of the textual content and reaction to classify vaccine stances. The architecture of our model is illustrated in Fig 5. Our model includes three components: a text encoder, reaction encoder, and classifier.

The text encoder induces linguistic features from tweet text. We fine-tuned a pre-trained Bidirectional Encoder Representations from Transformers (BERT) [25] on our target task. For each tweet, we carried out basic preprocessing, such as full-width half-width character conversion, case conversion, and removal of various symbols in the input before inputting it to BERT. To gain better classification performance, we conducted domain-adaptive pre-training (DAPT) [32] that continues pre-training for the pre-trained BERT with a corpus of a target

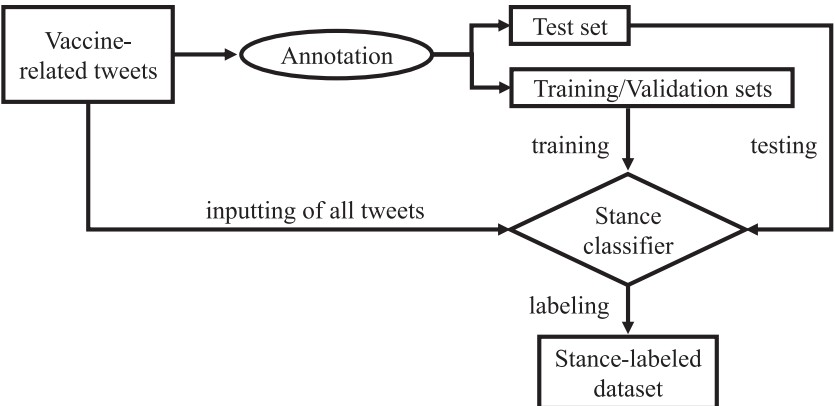

**Fig 4. An outline of our annotation, learning, and classification process.**

task domain. In our model, BERT was pre-trained with a masked language model (MLM) objective using our vaccine tweet dataset that we constructed in the previous section.

Motivated by the fact that a user's stance can be influenced by whom that user interacts with, the reaction encoder extracts reactions (retweets, quotes, and replies) between users to generate a reaction vector (RA vector) for each user representing who reacted to that user and whom the user reacted to. To reduce the computational costs, we used reactions to the most influential users. Specifically, we divided each month into three periods, 1st day to 10th, 11th to 20th, and 21st to 30th (31st), and we collected the top-10K users who reacted to others (hereinafter, information spreaders) and the top-10K users who others reacted to (hereinafter, information senders) for each period. We then vectorized the number of reactions between each user and top information spreaders/senders in the last three periods. The obtained RA vector was input to a fully-connected layer and tanh function to reduce the number of dimensions. The classifier inputs a concatenation of the tweet-text and reaction-graph vectors to a fully-connected layer. It then passes the output to the softmax function to make a prediction.

### Experiments on vaccination stance classification

**Settings.** For learning our vaccination stance classifier, we prepared datasets for training, development, and testing. Table 2 displays the statistics of the dataset. We created training and development datasets using the 9250 annotated tweets. To address potential annotator bias resulting from variations in the number of labeled tweets, we extracted an equal number of 125 tweets from each annotator to construct the development dataset, while the remaining 8750 tweets constituted the training dataset. To construct a reliable test dataset, we used the 500 tweets labeled by the four annotators to measure the inter-annotator agreements. To arrive at the final labels for the test dataset, a majority voting approach was used. In cases where there were disagreements among the annotators, resolutions were reached through collaborative discussions by the annotators.

To implement the text encoder, we used the Japanese BERT pre-learning model released by NICT, Japan (https://alaginrc.nict.go.jp/nict-bert/index.html). This Japanese-version BERT was pre-trained on Japanese Wikipedia. We used NICT_BERT-base_JapaneseWikipedia_100K. We set the maximum number of tokens to 160. On this BERT model, we performed DAPT with our vaccine tweet dataset for two epochs because no improvement in the performance (macro-F1 on the development data) was obtained after three epochs. For the reaction

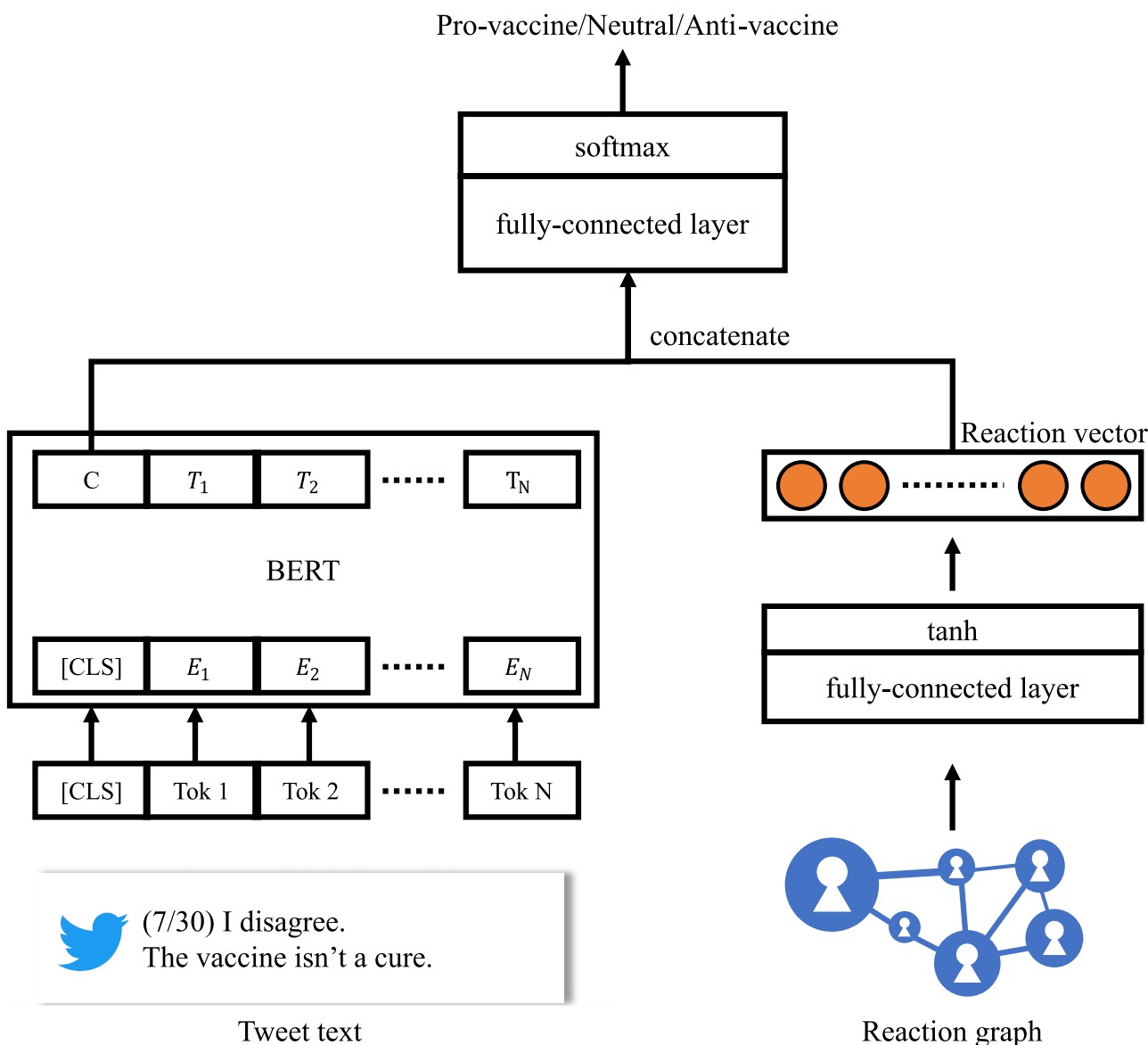

**Fig 5. Overview of our vaccine-stance classifier.**

encoder, we obtained RA vectors with 500 dimensions by feeding the original 95,016 dimensional vectors to two fully connected layers and the tanh function.

**Results.** To confirm the performance improvement of our classifier, we first compared the BERT classifier with that using DAPT. Subsequently, we incorporated the reaction encoder into the model to assess its efficacy. Table 3 lists the results including the precision, recall, $F_1$

**Table 2. Vaccination stance dataset.**

|  | Pro-vaccine | Neutral | Anti-vaccine |
|---|---|---|---|
| Training | 5237 | 3063 | 450 |
| Development | 392 | 96 | 12 |
| Test | 276 | 154 | 66 |

**Table 3. Comparison of performance of vaccination stance classifiers.**

| model | Pro-vaccine | | | Neutral | | | Anti-vaccine | | | Macro |
|---|---|---|---|---|---|---|---|---|---|---|
| | Prec. | Rec. | $F_1$ | Prec. | Rec. | $F_1$ | Prec. | Rec. | $F_1$ | $F_1$ |
| BERT | 0.875 | 0.837 | 0.856 | 0.698 | 0.766 | 0.731 | 0.462 | 0.375 | 0.414 | 0.667 |
| +DAPT | 0.887 | **0.884** | 0.886 | **0.773** | 0.773 | 0.773 | 0.471 | 0.500 | 0.485 | 0.714 |
| +DAPT+RAvec | **0.929** | 0.851 | **0.888** | 0.738 | **0.825** | **0.779** | **0.524** | **0.688** | **0.595** | **0.754** |

scores of each class, and macro-$F_1$ score. The BERT classifier with DAPT pre-training (+DAPT) consistently outperformed the original BERT classifier across all evaluation metrics. Notably, the recall of the anti-vaccine class exhibited significant improvement, highlighting the effectiveness of DAPT in enhancing the performance of the minority class. The classifier incorporating DAPT and the reaction encoder (+DAPT+RAvec) demonstrated the highest performance across most evaluation metrics. Similarly to DAPT, it significantly enhanced the performance of the anti-vaccine class, indicating its effective utilization of user interactions in the minority class.

Because the prediction performance of the anti-vaccine class was still not good due to the small number of tweets in the class, we set a probability threshold to obtain reliable labels when we applied it to our vaccine tweet dataset. When the class with the maximum output of the softmax function was the anti-vaccine class, we set a threshold of 0.7 to the output probability of the anti-vaccine class. Thus, the precision of the anti-vaccine class increased from 0.524 to 0.700, which is not that much worse than the other classes. Instead, the recall of the class decreased from 0.688 to 0.438.

## Analysis

We applied our vaccination stance classifier to all tweets in our vaccine tweet dataset. Using these automatically labeled tweets, we examined how users' stances on vaccination were formed and the information-sharing behaviors that potentially influenced this process. In all analyses, we have complied with the terms and conditions of Twitter (X Corp.) at https://developer.twitter.com/en/developer-terms/more-on-restricted-use-cases. The following results update our previous findings [18] by incorporating our improved vaccination stance classifier (BERT+DAPT+RAvec).

### Distribution of users' stances

Using all the tweets labeled with our vaccination stance classifier, we determined users' stances. We assumed that users do not change their stances in a short period of time and divided each month into three periods to aggregate tweet labels by each user using majority votes. In the case of a tie, we determined the user's stance with priority toward pro-, neutral, and anti-vaccine in alignment with the order of the prediction precision.

Fig 6 illustrates the distribution of users' stances in the 15 time periods during the 5 months. The number of pro-vaccine users was comparative with the neutral users during the first two periods. As the vaccination campaign progressed, the number of pro-vaccine users gradually increased. Starting in September 2021, there was decline in the number of pro-vaccine users coinciding with the achievement of approximately 50% full vaccination coverage, suggesting that interest in vaccination may have waned among these users around that time. The number of anti-vaccine users remained consistently small compared to other stances across all periods,

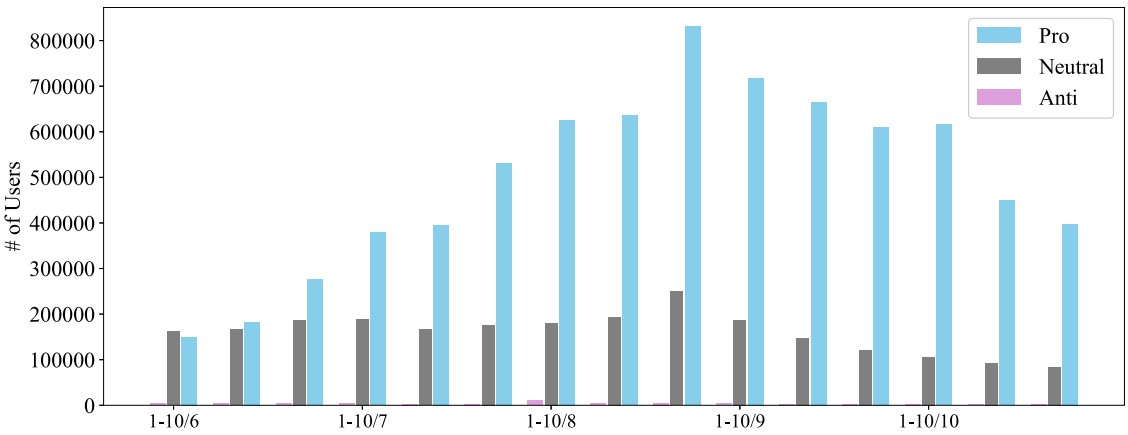

**Fig 6. Changes in number of users with each stance.**

suggesting that their influence was minimal, but their interest in the vaccination campaign remained unchanged.

## Transition in polarization between vaccination stances

To examine the polarization between pro- and anti-vaccine users that is reported in online debates on vaccines for other infectious diseases [20, 33, 34], we created a graph of interactions between users, depicting the distribution of user stances based on a prior study [20]. We used undirected retweet graphs, where each node represented a user, and each edge represented the presence of retweets between users at both ends. To observe the changes in polarization over-time, we constructed retweet graphs for the 15 time periods and visualized each graph using Gephi (https://gephi.org/), a graph visualization tool. We only depicted nodes with degrees of 30 or higher to focus on users actively sharing information.

Fig 7 shows the retweet graphs for the first ten days of each month. The node colors show users' vaccination stances, and the numbers show how many users have each stance. We can see densely connected nodes comprising three groups in each period, which is associated with the three different stances. There were relatively sparser connections between the pro- and anti-vaccine stances than those between the neutral group and the other two. This observation suggests a persistent polarization between the pro- and anti-vaccine groups throughout the periods.

We estimated the degree of polarization for each period and examined its transition. For this estimation, we used a modified version of the Random Walk Controversy (RWC) [29], one of the most common measures for estimating the degree of polarization between two communities in a graph. We extended this method to measure the polarization between pro- and anti-vaccine communities in the presence of a third, neutral community. Our modified RWC first identified densely connected nodes within the retweet graph as communities. The METIS algorithm, which was used in the original RWC, is designed to divide a given graph into $k$ clusters with an equal number of vertices. However, in our case, pro- and anti-vaccine communities has different number of vertices, making the METIS algorithm less suitable. Therefore, we used the Louvain method [30], which produced a better partitioning result in terms of modularity better reflecting the structure of the retweet graph. In this method, the "resolution" parameter has a direct impact on the minimum size of the communities, which we set at 2. We then assigned one of three stances to each of the

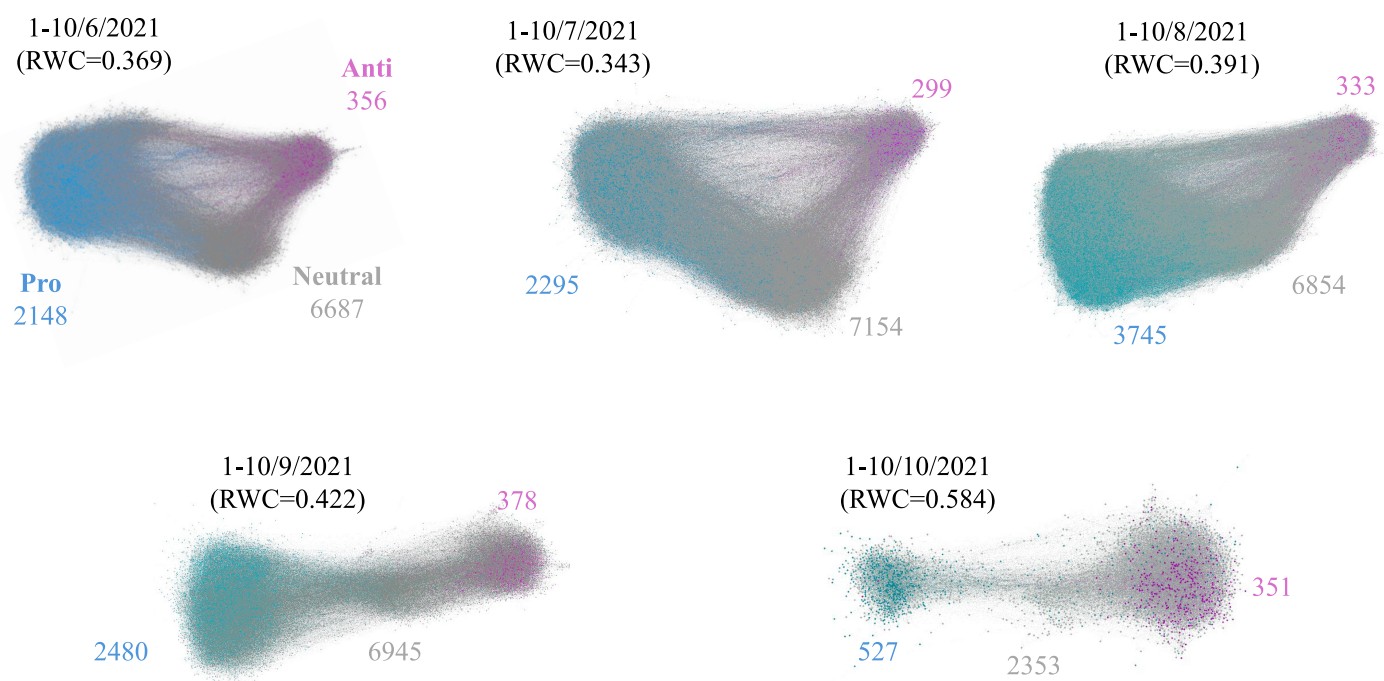

**Fig 7. Evolution of polarization of reaction graphs, RWC, and number of users with each stance.**

three largest communities through a majority vote among the community's users. Subsequently, we computed the RWC between the pro- and anti- communities for each period. RWC was originally designed to measure polarization between two communities by calculating the ratio of random walks starting from either community that remain within the same community. Our modified version just ignored the neutral community by starting the random walk from a node of either the pro- or anti-vaccine community and finishing it at $k$ highest-degree nodes within either of pro- or anti-vaccine community. We used $k = 10$ highest-degree nodes as the goal nodes.

As shown in Fig 7, the three groups were initially segregated by their stances. However, the neutral and pro-vaccine groups became densely connected over the first three months. After the beginning of September, these two groups rapidly decreased in size, coinciding with a rapid increase in the number of vaccinated people, while the size of the anti-vaccine group remained consistent. This trend aligns with the decreasing number of vaccine-related tweets since late August, when first-dose vaccination coverage in Japan exceeded 58%, as shown in Fig 2.

The increase in RWC of the retweet graphs can be attributed to two main factors: the diminishing number of direct edges between pro- and anti-vaccine users, and a notable decrease in the number of neutral users bridging these opposing stances over time. These observations indicate that neutral users primarily communicated with pro-vaccine users, and both groups lost interest or stopped posting after receiving their first dose. The influence of anti-vaccine users remained limited in Japan, although their activities continued. To determine whether these anti-vaccine users were bots or a special type of user, we investigated 50 randomly sampled anti-vaccine accounts. We did not find any common characteristics among these users, nor did we find any bot accounts.

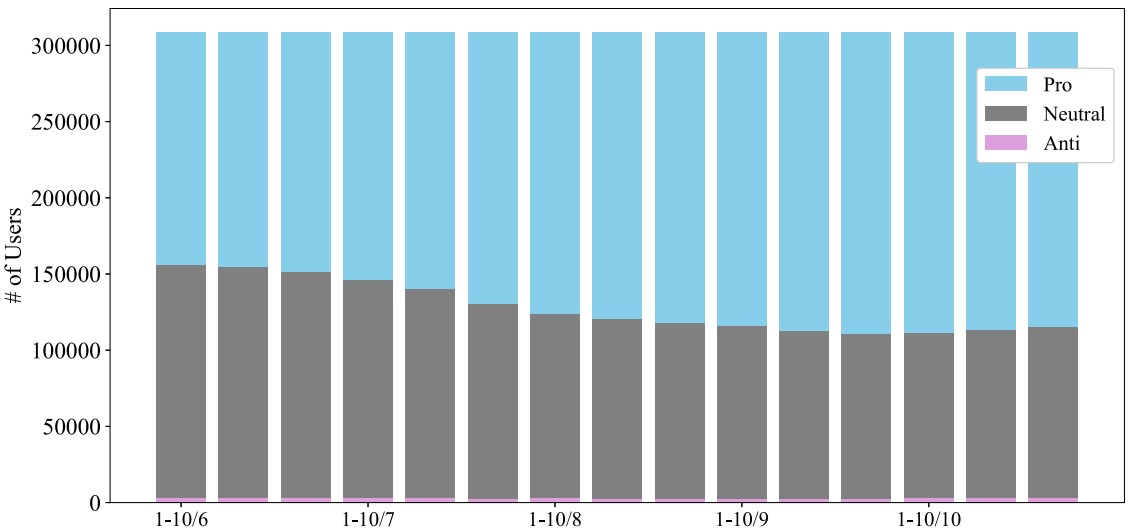

**Fig 8. Changes in user stance distribution.**

## Stance formations of users

We examined the stance formations of active users who consistently posted during the data collection period. Firstly, we identified 308,789 users who posted in at least 4 months out of 5 months of data collection window. As defined in the analyses above, each month is divided into three periods. A user's stance during each period is determined by the majority vote of stance labels from their tweets posted within that period. If a user did not post during a particular period, their stance was assigned based on their last known stance from a previous period with posts or their first stance in subsequent periods if they had no prior posts. Finally, we conducted analyses on the formation of users' stances and their information-sharing behaviors over time. Throughout and after the analyses, we ensured that no information identifying individual users was accessed.

We first illustrate the transition in users' stance distribution over 5 months (15 periods) in Fig 8. In the initial period, the number of pro-vaccine and neutral users was nearly equal, with very few anti-vaccine users. The number of pro-vaccine users steadily increased over subsequent periods, while the number of anti-vaccine users remained consistently small.

To further investigate the exchange of users between stance groups, we present the matrix showing the number of users transitioning between stances from the initial to the final periods in Table 4. As depicted, both the pro-vaccine and anti-vaccine groups predominantly exchanged members with the neutral group, and the exchanges between the pro-vaccine and anti-vaccine groups are notably smaller. The stance of pro-vaccine users was relatively stable,

**Table 4. Transition matrix between users' initial stances and final stances.**

| | | To | | | Total |
|---|---|---|---|---|---|
| | | **Pro-vaccine** | **Neutral** | **Anti-vaccine** | |
| From | Pro-vaccine | 110,817 | 42,036 | 557 | 153,410 |
| | Neutral | 82,204 | 68,256 | 1989 | 152,449 |
| | Anti-vaccine | 851 | 1608 | 471 | 2930 |
| | Total | 193,872 | 111,900 | 3017 | 308,789 |

whereas that of anti-vaccine users was more vulnerable. Among the 153,410 initially pro-vaccine users, 110,817 (73%) maintained their stance, while nearly all of the remaining users shifted to a neutral stance. Among the 2,930 initially anti-vaccine users, only 471 (16%) maintained their stance, while 1,608 (55%) shifted to a neutral stance, and a relatively small number, 851 (29%), shifted to pro-vaccine. The results show that both the pro-vaccine and anti-vaccine groups primarily exchanged members with the neutral group, and less with each other. These findings align with the sparse connections observed between pro- and anti-vaccine users, as shown in Fig 7. Therefore, our primary focus is on the stance formation of the 152,449 initially neutral users. Of these, 82,204 (54%) shifted to pro-vaccine, which significantly outweighed the shift from pro-vaccine to neutral. In contrast, only 1,989 (1.3%) shifted to anti-vaccine, a shift nearly equal to the movement from anti-vaccine to neutral.

## Information-sharing behaviors associated with stance formation changing stances

As shown above, since there were fewer transitions between the pro- and anti-vaccine groups, we focused on users who were initially neutral, changed their stance only once to either pro-vaccine (neutral-to-pro) or anti-vaccine (neutral-to-anti), and maintained that stance until the last period. For each month, we compared the information-sharing behaviors of users who changed their stance that month with those who remained neutral (remaining-neutral) during that month. In our dataset, among the 152,449 users who were initially neutral in the first period, 60,289 users fell into one of three categories: neutral-to-pro, neutral-to-anti, or remaining-neutral. Among these, 42,905 changed to pro-vaccine once, 603 changed to anti-vaccine once, and both groups maintained that stance until the last period. The remaining 16,781 users stayed neutral throughout all the periods. We analyzed their information-sharing behaviors, focusing specifically on interactions with other user accounts and external sites.

**What kinds of users were referred to by users who formed their stances?.** We investigated which user accounts were most referred to (replied, retweeted, or quoted) by either the neutral-to-pro or neutral-to-anti users during their stance formation. These referred accounts were then compared to those referred to by the remaining-neutral users. For each month, we classified users into three groups: neutral-to-pro, neutral-to-anti, and remaining-neutral, based on the formation of their stances during that month. For each group, we extracted the user accounts referred to by members of each group during the period when their stance changed, as well as during the three periods prior to the change. If one user referred to the same user account multiple times in the month, it was counted only once. After identifying the user accounts frequently referred to by each group, we classified these accounts by their attributes, such as their titles, jobs and belonging organizations. The authors assigned these attributes to each user accounts based on their profiles, several recent tweets. Table 5 provides the definition of each attribute.

Fig 9 shows the attributes of the top 30 user accounts most referred to by the neutral-to-pro, remaining-neutral, and neutral-to-anti users in each month. The neutral-to-pro and remaining-neutral users consistently referred to a notable number of user accounts belonging to medical workers. In contrast, the neutral-to-anti users were unlikely to refer to such accounts. In Japan, many medical doctors voluntarily provided information on the effects and risks of vaccines to the public through their personal accounts, which may have played a important role in increasing vaccination coverage. We also observed that users referred to a wider variety of user accounts when they changed their stances. As shown in Fig 9, neutral-to-pro and neutral-to-anti users referred to a diverse range of user accounts, such as artists, business persons, professional writers and influencers, compared to remaining-neutral users.

**Table 5. Attributes of referred user accounts.**

| Labels | Definition |
|---|---|
| Government | Official accounts of government agencies |
| Politician | Diet members, prefectural and city council members |
| Medical worker | Doctors, nurses, and health professionals |
| Researcher | Researchers from universities, research institutes, and companies |
| Mass media | TVs, newspapers, and magazines |
| Web news | Official accounts of web news sites |
| Business person | White-collar workers, self-employed persons, and entrepreneurs |
| Influencer | Influential accounts on YouTube, Instagram, etc. |
| Professional writer | Novelists and journalists |
| Artist | Illustrators and comic writers |
| Uncategorized | Users with other attributes |
| No info | Users with no information regarding attributes |

To further examine user accounts exclusively referred to by either the neutral-to-pro or neutral-to-anti users in comparison with remaining-neutral users, we carried out a chi-squared test of independence on two user account groups at a significance level of 5%. Among the user accounts that passed the chi-squared test, we extracted the top 30 user accounts most frequently referred to by users in each group and assigned attributes in the same manner.

Fig 10 shows the attributes of the top 30 user accounts referred to by neutral-to-pro and neutral-to-anti users over time. The neutral-to-pro users tend to refer to the vaccination experiences of a diverse group of users. They notably referred to many artist accounts that shared their vaccination experiences through comics and illustrations. Tweets featuring comics or illustrations tend to attract more attention than text-only tweets, and such tweets may have played a significant role.

References to office workers and professional writers were consistently shown. They shared information about the vaccination procedure at venues and their personal experiences with vaccine side effects. In October, there was a sharp increase in references to medical workers. This was due to a combination of factors, including mentions of the downsizing of vaccination venues, information about vaccines provided by the Ministry of Health, Labour and Welfare and The Japanese Society for Vaccinology, and reports on studies concerning vaccinations for children.

It was found that the neutral-to-anti users tended to refer to user accounts who highlighted the negative aspects of vaccines. They mainly emphasized the dangers and ineffectiveness of vaccines, or criticize policies related to vaccines, such as opposing vaccine certificates. It is also noteworthy that we were unable to identify the occupations for about half of the accounts referred to by the neutral-to-anti users.

**What types of external sites were shared by users who determined their stances?.**   We next investigated the external sites that were shared by the neutral-to-pro and neutral-to-anti users during their stance formation. Similarly to the analysis of referred user accounts, we classified users into three groups (neutral-to-pro, remaining-neutral, and neutral-to-anti) for each month. For each group, we extracted members' tweets and retweets containing links to external sites during the period when their stance changed, as well as during the three periods prior to the change. If one user shared the same link multiple, it was counted only once. After identifying the external links frequently shared by each group, the authors assigned categories to each link based on the types of their web sites. Table 6 provides the definition of each category.

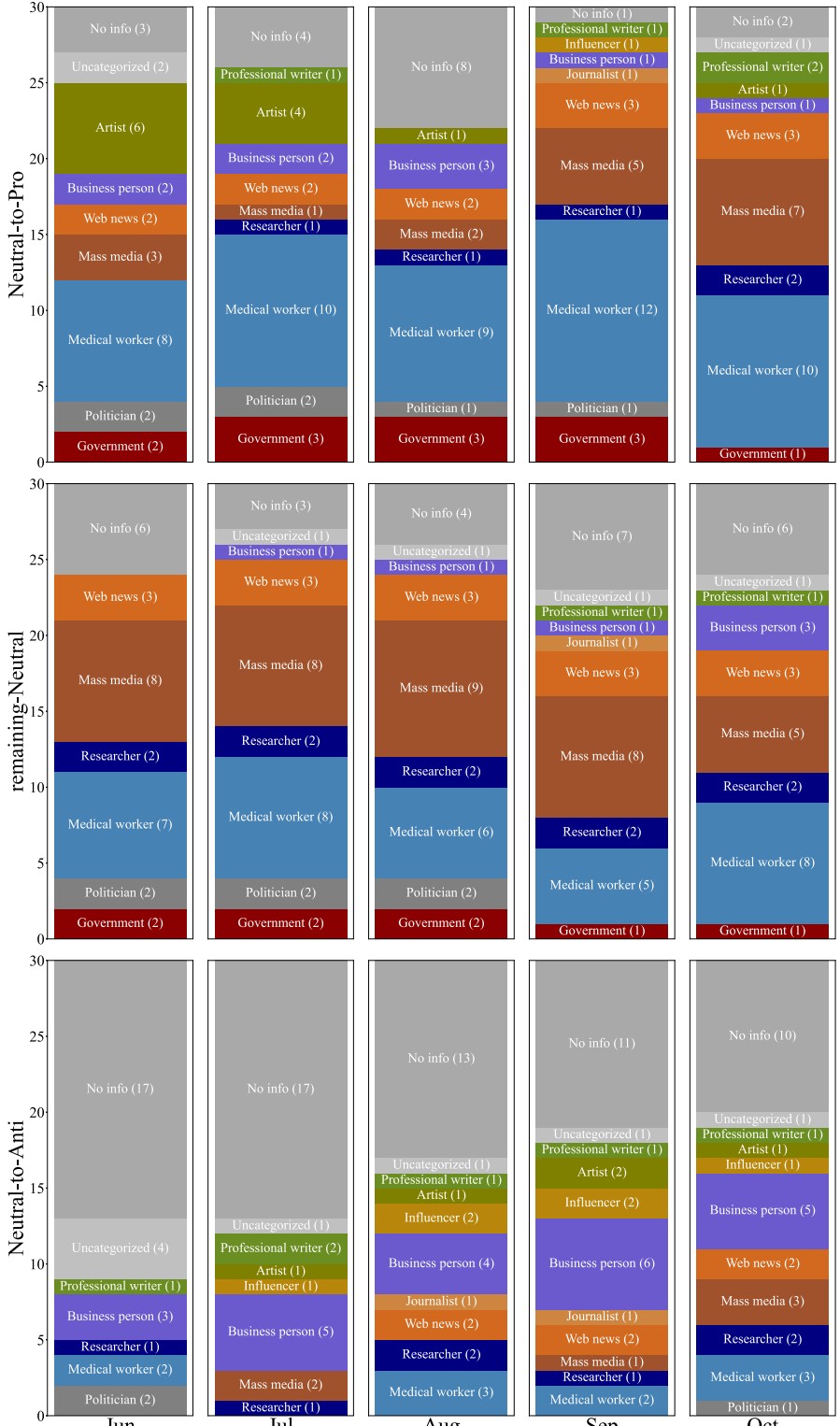

**Fig 9. Users who were most referred to by neutral-to-pro users (top), remaining-neutral users (middle), and neutral-to-anti users (bottom).**

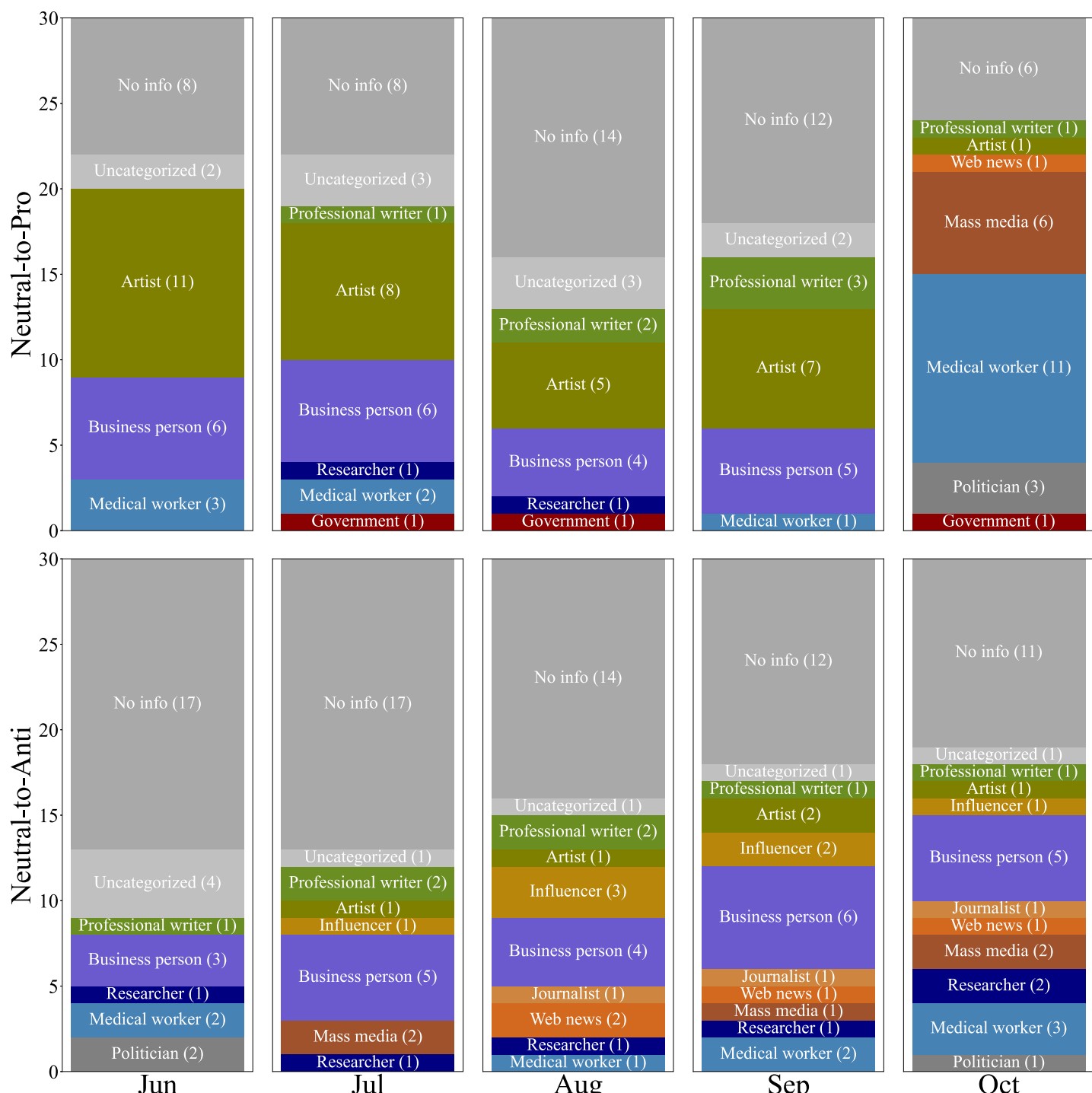

**Fig 10. Users, who passed the chi-squared test of independence, referred to by neutral-to-pro users (top) and neutral-to-anti users (bottom).**

Fig 11 shows the categories of the top 30 external sites referred to by the neutral-to-pro, remaining-neutral, and neutral-to-anti users for five months. The neutral-to-pro users and remaining-neutral users mainly shared links to mass media or web news sites, while the neutral-to-anti users shared alternative information sources. There was no significant difference

**Table 6. Categories of shared external sites.**

| Labels | Criteria | Example |
|---|---|---|
| Government | Government agencies | www.mhlw.go.jp, corona.go.jp |
| Politician | Politicians' personal sites | www.taro.org |
| Healthcare institution | Hospitals and medical institutions | trihealth.com, www.hosp.ncgm.go.jp |
| Pharmaceutical | Companies developing and manufacturing medications | www.pfizer.com |
| Academic organization | Universities, academic associations, and journals | www.nature.com, www.thelancet.com, www.hc.u-tokyo.ac.jp |
| Mass media | TVs, newspapers, and magazines | www.cnn.com, www.fnn.jp |
| COVID-info | Web sites curating information regarding COVID-19 | covid-vaccine.jp, www.covid-datascience.com |
| Web news | News sites exclusively through online channels | news.yahoo.co.jp, news.livedoor.com |
| SNS | Social media sites (except for Twitter) | www.facebook.com, www.instagram.com |
| Video hosting | Video hosting sites | www.youtube.com, odysee.com, rumble.com |
| BBS, Blog | BBSs and personal blogs | 5ch.net, togetter.com |
| Uncategorized | Other miscellaneous sites | detail.chiebukuro.yahoo.co.jp, ja.wikipedia.org |

between neutral-to-pro users and remaining-neutral users in this figure. Neutral-to-pro users tended to refer slightly more to web news sites, while there were no significant differences in the referenced articles because these sites predominantly reported articles already covered by mass media. This indicates a tendency for neutral-to-pro users to access such articles via web news sites rather than directly through mass media sites. The neutral-to-anti users frequently shared video hosting sites, BBS, and blog sites, which means they prefer these alternative information sources over mass media sites such as TV or newspapers. For example, in July, the number of video hosting sites was ten in Fig 11. Among these, the number of major video sharing sites was only four, while the rest six sites were relatively minor video sharing sites. These ten videos included titles such as "Vaccines Cause Tragedies" and others claiming that vaccination for children poses a high risk, indicating that video sharing platforms have become mediums for the spread of anti-vaccine sentiments. Similarly, blogs and BBSs criticizing the government's stance on vaccines or detailing post-vaccination deaths were frequently referenced. Specifically, these included blogs claiming that it is healthier not to get vaccinated, BBSs introducing cases of people who died after vaccination, and blogs reporting that the government concealed the number of deaths caused by vaccines. Among these, blogs that particularly highlight the dangers of vaccines often lack evidence or rely on suspicious evidence, supporting the hypothesis that blogs and BBSs serve as a hotbed of anti-vaccine.

To further examine external links exclusively referred to by either the neutral-to-pro or neutral-to-anti users in comparison with remaining-neutral users, we carried out a chi-squared test of independence on two sets of shared links at a significance level of 5%. Among the shared links that passed the chi-squared test, we extracted the top 30 links most frequently referred to by users in each group and assigned categories in the same manner.

Fig 12 shows the top30 frequently shared external websites by the neutral-to-pro and -anti users over five months. The top of Fig 12 shows sites shared by the neutral-to-pro users. The majority of these were web news sites, with a much smaller number being mass media sites. However, most of the shared news articles were also covered by mass media, and we did not find significant differences in content compared with Fig 11. The bottom of Fig 12 shows sites shared by the neutral-to-anti users. Similarly, we did not find significant differences compared with Fig 11.

Fig 13 illustrates word cloud created from the headlines of the links shared by the neutral-to-pro and -anti users over time. We obtained the headlines from the external sites which were

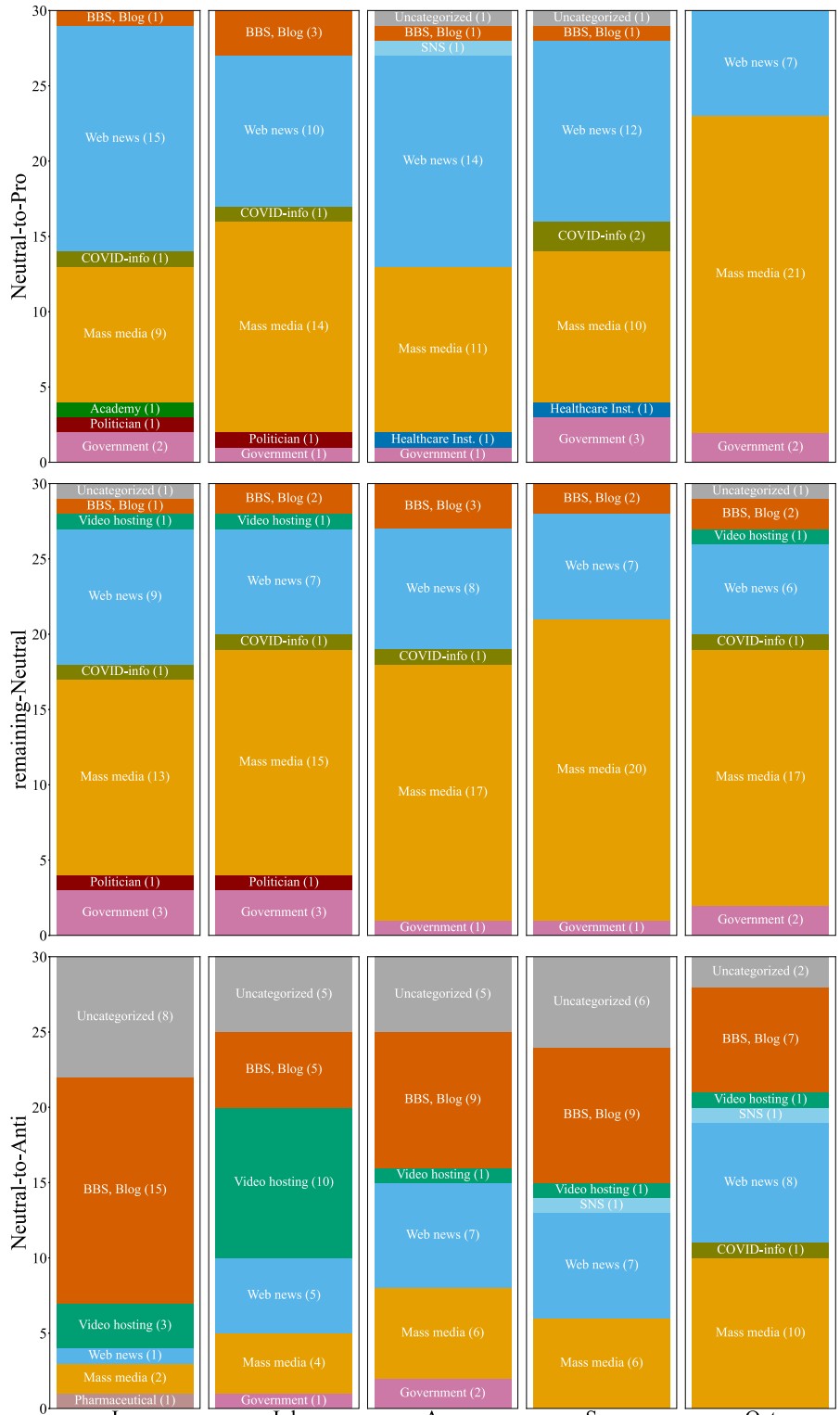

**Fig 11. External sites which most shared by neutral-to-pro users (top), remaining-neutral users (middle), and neutral-to-anti users (bottom).**

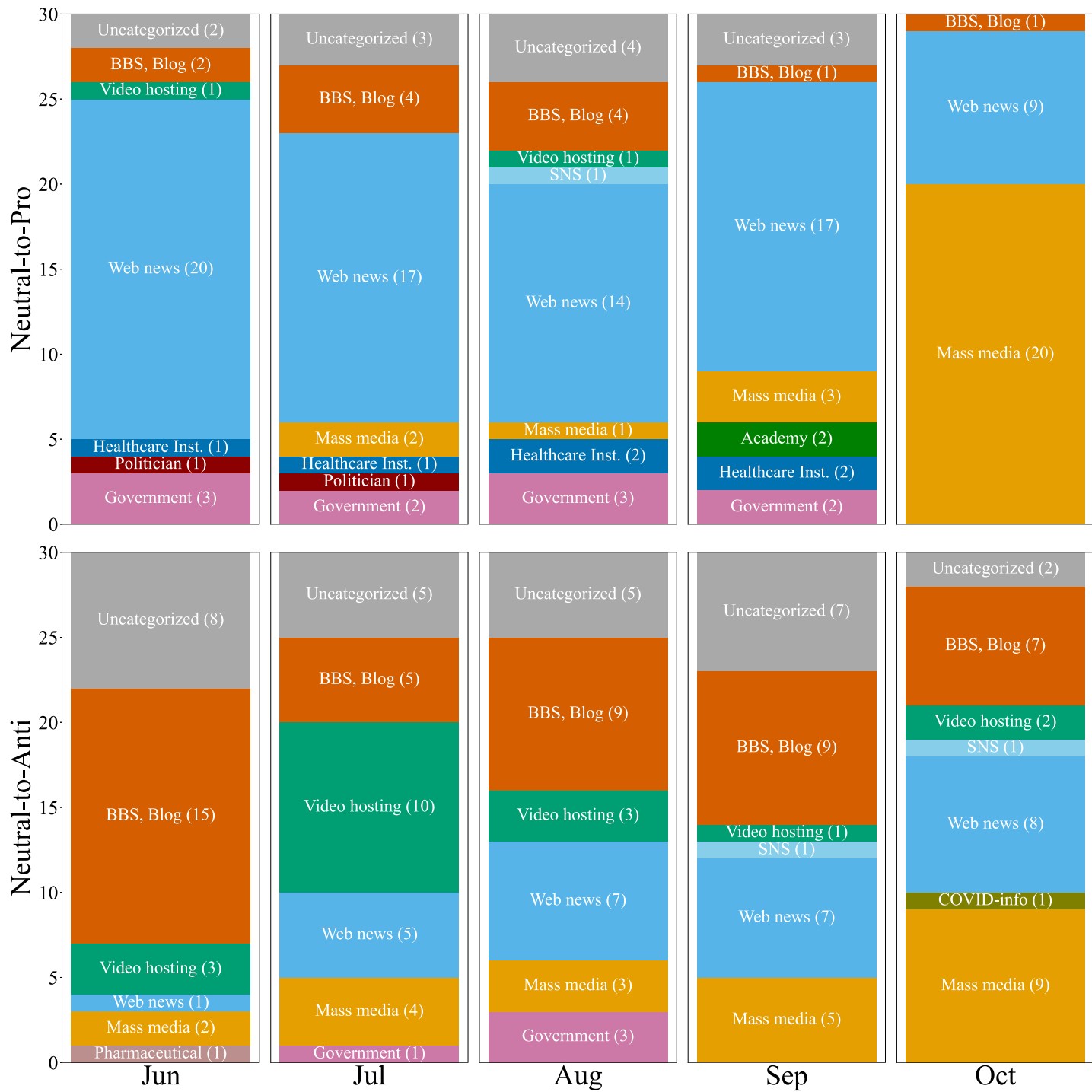

**Fig 12. External sites which passed the chi-squared test, shared by neutral-to-pro users (top) and neutral-to-anti users (bottom).**

significantly frequently shared by neutral-to-pro or -anti users respectively at a significance level of 5%. From the shared sites, we extracted nouns frequently used in the titles of the headlines using MeCab (https://taku910.github.io/mecab), one of the most popular Japanese tokenizers. We performed a chi-squared test of independence on two word groups used in the

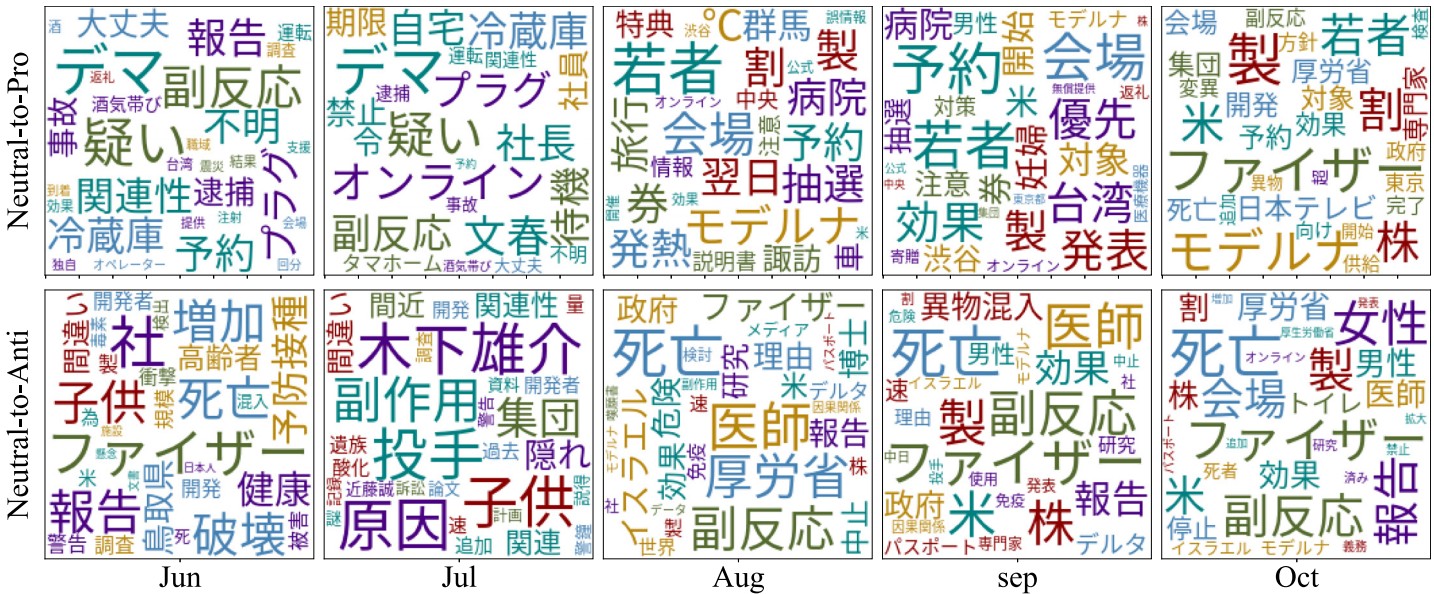

**Fig 13. Changes in keywords in titles of external sites referred to by neutral-to-pro users (top) and neutral-to-anti users (bottom).**

titles shared by the neutral-to-pro (or neutral-to-anti) users and the remaining-neutral users at a significance level of 5%. When one user shared several sites and there was an overlap of tokens between the titles of the sites, the token was counted only once. After identifying the typical words used in each group, we conducted a deeper analysis of the contents containing these keywords.

For neutral-to-pro users, we observed a significant interest in websites related to vaccine reservations. Specifically, in all months except for July, we identified an increased usage of the word "reservation" as users checked and shared information on how to make a vaccine reservation or tweeted that they had successfully made one.

In the first few months, neutral-to-pro users displayed significant interest in the potential drawbacks of vaccination and tended to refer to authoritative sources such as government agencies and officials in charge of vaccinations. During this period, Taro Kono, the Cabinet minister in charge of vaccinations, issued warnings about vaccine-related false rumors on his official website, which led to the emergence of the term "false rumor" in June and July (https://www.taro.org/2021/06/%e3%83%af%e3%82%af%e3%83%81%e3%83%b3%e3%83%87%e3%83%9e%e3%81%ab%e3%81%a4%e3%81%84%e3%81%a6.php). Additionally, the Ministry of Health, Labour and Welfare reported cases of suspected vaccine side effects online (https://www.mhlw.go.jp/stf/seisakunitsuite/bunya/vaccine_hukuhannou-utagai-houkoku.html), leading to the appearance of words such as "suspicion," "side effects," and "report."

Starting in August, the vaccination campaign for young people began and attracted significant attention from neutral-to-pro users for several months. In August, news reports covered the opening of vaccination venues for young people (https://news.livedoor.com/article/detail/20718403, https://news.livedoor.com/lite/article_detail/20769771), leading to the appearance of "young people," "venue," "coupons," and "reservation." Some of these words continued to appear in the following months.

Since August, there has been increased interest in the effectiveness and safety of vaccines, confirmed by authorities. The word "Moderna" appeared from August to October. In August, news reported that the Ministry of Health, Labour and Welfare stated that over 80% of people

who received the Moderna vaccine experienced a fever and recommended resting the day after vaccination (https://news.livedoor.com/article/detail/20649919). Attention to Moderna dropped in September but rose again in October when it was reported that some Moderna vaccines contained stainless steel contaminants (https://www.tokyo-np.co.jp/article/130189). In October, the word "Pfizer" also appeared. This was due to news that 80% of the antibodies from the Pfizer vaccine decreased within six months (https://times.abema.tv/news-article/8673379) and that a booster shot had an effectiveness of 96% (https://news.yahoo.co.jp/articles/04f1fe37b1a5f7a66ecee6081de50018c102c559).

As shown in bottom of Fig 13, the word cloud of the neutral-to-anti users contained words exaggerating the negative aspects of the vaccine, such as "death," "side effects", and "danger." This suggests that the neutral-to-anti users were particularly anxious about the vaccine's safety. Furthermore, BBSs and blog articles that report foreign information related to vaccines, particularly highlighting their drawbacks, attracted significant attention among neutral-to-anti vaccine users. For instance, in August, the term "Israel" began to appear, due to two blog articles: one claiming that Pfizer and Israel agreed to conceal the side effects of the vaccine (https://tocana.jp/2021/08/post_218206_entry.html), and another warning that the number of positive COVID-19 cases in Israel did not decrease despite booster shots, contrary to reports of their effectiveness (https://johosokuhou.com/2021/09/29/51798/). Since Israel was one of the first countries to start vaccinations globally, negative information about vaccines related to Israel attracted considerable attention from the neutral-to-anti users. Similarly, since September, the term "US" has also appeared, as the country was another early adopter of vaccination. This term emerged from blog articles claiming that 60% of American doctors refused to get vaccinated (http://www.rui.jp/ruinet.html?i=200&c=600&t=6&k=0&m=368469&g=131203) and introducing research from the US suggesting that vaccines posed a high risk are dangerous for young people (https://note.com/you3_jp/n/n463d19aeaf03). This appearance is considered to be for similar reasons as that of Israel.

It was also found that doctors and medical experts might be influencing neutral-to-anti vaccine users towards anti-vaccine sentiments. The results also suggest that anti-vaccine doctors and experts may be pushing people towards anti-vaccine sentiments. In August, the words "doctors" appeared, referring to the news about a group of 450 doctors and legislators submitting a petition to suspend vaccinations (https://www.sanspo.com/article/20210624-IOQJULJCVRMBXMZXIDJG6SDUHA). Additionally, in June and July, the word "developer" appeared, due to a blog article about a person claiming to be a vaccine developer who warned that the vaccine is poison (http://blog.nihon-syakai.net/blog/2021/06/12371.html?g=132207). This was actually an opinion by a Canadian vaccine researcher who had seen internal Pfizer documents. The appearance of the word "developer" over two months suggests that its influence was significant. Such news, based on the views of doctors and experts, has also garnered considerable attention, suggesting that even users with anti-vaccine tendencies find statements from individuals in authoritative medical positions to be effective. This is consistent with neutral-to-anti users tending to refer to accounts from medical workers and researchers shown in Figs 9 and 10.

## Conclusion

To draw a lesson from the successful COVID-19 vaccination campaign in Japan, we analyzed stance formations of Twitter users towards COVID-19 vaccination. We developed a BERT-based stance classifier with reaction information and applied it to all vaccine-related tweets posted from June to October 2021.

Analysis of the distribution of stances and the polarization of the pro-vaccine and anti-vaccine users revealed that the number of pro-vaccine users greatly exceeds the number of anti-vaccine users, and interactions among them are relatively sparse. The number of pro-vaccine users increased from June when the vaccination campaign for individuals aged 18 to 64 started, but it decreased from September when the vaccination coverage reached around 50%. The impact of anti-vaccine users was relatively insignificant, as their numbers remained consistently low throughout the analyzed period. However, it is noteworthy that their level of interest in the vaccination campaign remained unchanged. We found that polarization between the pro-vaccine and anti-vaccine users increased over time. Additionally, we observed that neutral users tended to react more frequently to the pro-vaccine users rather than the anti-vaccine users. These findings may explain why the majority of neutral users who became other stances shifted towards being pro-vaccine. It underscores the significance of providing reliable and timely information to neutral users to effectively improve the vaccination coverage.

Users who transitioned to the pro-vaccine stance often relied on traditional and authoritative information sources such as medical doctors, mass media, governments, and politicians. This indicates that the information provided by these sources played a crucial role in influencing individuals' decision to get vaccinated. Notably, many medical doctors took the initiative to share updated vaccine information through their personal accounts, highlighting the significant impact of personal activities in improving vaccination coverage.

Such users were also found to frequently refer to the vaccination experiences of a diverse group of user accounts. This suggests that users considering vaccination are preparing by looking at these experiences to smoothly undergo vaccination. Notably, they referred to many artist accounts that shared their vaccination experiences through comics and illustrations.

In contrast, users who transitioned to the anti-vaccine stance often relied on alternative information sources such as BBSs, blogs, video sharing sites, and accounts with unknown occupations. A word cloud analysis of the neutral-to-anti users revealed a predominant interest in vaccine safety. While it is natural for individuals to have concerns about vaccine risks, these users exhibited an elevated level of worry that led them to trust suspicious information, including rumors, gossip, and fake news. The information found on these platforms often lacked moderation and contributed to the dissemination of misinformation.

These findings highlight the importance of disseminating information through diverse channels to prevent neutral users from encountering misinformation in alternative information sources. By ensuring the availability of accurate and reliable information across various platforms, we can help neutral users make informed decisions and avoid being influenced by misleading content.

One limitation of our study is that we categorized both 'vaccine acceptance' and 'vaccine uptake' under pro-vaccine, despite their distinct nature. Expressing acceptance of COVID-19 vaccination does not necessarily mean actual uptake, and vice versa. A systematic review and meta-analysis revealed a significant difference between global acceptance rate of COVID-19 vaccination (67.8%) and its uptake rate (42.3%) [35]. In our annotation dataset, almost all of pro-vaccine posts reported uptake (95.7%), posing a challenge for analyzing these two concepts separately.

In future studies, interpretable models could reveal the key features determining the classification of tweets into different stances. For instance, these models could identify common words or influencers associated with each stance. For example, employing such models could provide popular words or popular influencers associated with each stance. The application of NLP techniques could deepen our understanding. A thematic analysis such as LDA could unveil prevalent topics among the neutral-to-pro/anti users. Emotion fusion could offer

insight into emotional features associated with each stance, and their effect on stance formation.

## Author Contributions

**Conceptualization:** Sho Cho, Shohei Hisamitsu.

**Data curation:** Sho Cho, Shohei Hisamitsu.

**Formal analysis:** Sho Cho, Shohei Hisamitsu.

**Investigation:** Sho Cho, Shohei Hisamitsu, Hongshan Jin.

**Methodology:** Sho Cho, Shohei Hisamitsu.

**Project administration:** Masashi Toyoda, Naoki Yoshinaga.

**Resources:** Masashi Toyoda.

**Software:** Sho Cho, Naoki Yoshinaga.

**Supervision:** Hongshan Jin, Masashi Toyoda, Naoki Yoshinaga.

**Validation:** Sho Cho.

**Visualization:** Sho Cho, Shohei Hisamitsu.

**Writing – original draft:** Sho Cho, Hongshan Jin, Masashi Toyoda, Naoki Yoshinaga.

**Writing – review & editing:** Sho Cho, Hongshan Jin, Masashi Toyoda, Naoki Yoshinaga.

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
