## [Decision Letter · Decision Letter 0]

1 Sep 2023

PONE-D-23-22865Analyzing information sharing behaviors during stance shifts on COVID-19 vaccination among Japanese Twitter usersPLOS ONE

Dear Dr. Cho,

Thank you for submitting your manuscript to PLOS ONE. After careful consideration, we feel that it has merit but does not fully meet PLOS ONE’s publication criteria as it currently stands. Therefore, we invite you to submit a revised version of the manuscript that addresses the points raised during the review process.

ACADEMIC EDITOR:

Kindly address the comments made by reviewers.

We look forward to receiving your revised manuscript.

Kind regards,

Ankit Gupta

Academic Editor

PLOS ONE

2. In your Methods section, please include additional information about your dataset and ensure that you have included a statement specifying whether the collection and analysis method complied with the terms and conditions for the source of the data.

https://ieeexplore.ieee.org/document/10068695

In your revision ensure you cite all your sources (including your own works), and quote or rephrase any duplicated text outside the methods section. Further consideration is dependent on these concerns being addressed.

4. Please note that PLOS ONE has specific guidelines on code sharing for submissions in which author-generated code underpins the findings in the manuscript. In these cases, all author-generated code must be made available without restrictions upon publication of the work. Please review our guidelines at https://journals.plos.org/plosone/s/materials-and-software-sharing#loc-sharing-code and ensure that your code is shared in a way that follows best practice and facilitates reproducibility and reuse.

“This research was conducted as part of “COVID-19 AI & Simulation Project” run by Mitsubishi Research Institute commissioned by Cabinet Secretariat, JAPAN. The methods for analysis were developed with support from JST CREST Grant Number JPMJCR19A4 and JSPS KAKENHI Grant Number JP21H03445.”

“This research was conducted as part of "COVID-19 AI & Simulation Project" run by Mitsubishi Research Institute commissioned by Cabinet Secretariat, JAPAN. The methods for analysis were developed with support from JST CREST Grant Number JPMJCR19A4 and JSPS KAKENHI Grant Number JP21H03445.

6. We noted in your submission details that a portion of your manuscript may have been presented or published elsewhere. [This paper is an extended version of our paper published in Diachronic Analysis of Users' Stances on COVID-19 Vaccination in Japan using Twitter, Proceedings of the 2022 IEEE/ACM International Conference on Advances in Social Networks Analysis and Mining (ASONAM), 2022, pp. 237-241. We have improved our classifier for stance detection, updated all experimental results, and added some new results.] Please clarify whether this publication was peer-reviewed and formally published. If this work was previously peer-reviewed and published, in the cover letter please provide the reason that this work does not constitute dual publication and should be included in the current manuscript.

7. We note that you have stated that you will provide repository information for your data at acceptance. Should your manuscript be accepted for publication, we will hold it until you provide the relevant accession numbers or DOIs necessary to access your data. If you wish to make changes to your Data Availability statement, please describe these changes in your cover letter and we will update your Data Availability statement to reflect the information you provide.

8. We note that Figure 8 in your submission contain copyrighted images. All PLOS content is published under the Creative Commons Attribution License (CC BY 4.0), which means that the manuscript, images, and Supporting Information files will be freely available online, and any third party is permitted to access, download, copy, distribute, and use these materials in any way, even commercially, with proper attribution. For more information, see our copyright guidelines: http://journals.plos.org/plosone/s/licenses-and-copyright.

1. You may seek permission from the original copyright holder of Figure 8 to publish the content specifically under the CC BY 4.0 license.

Reviewers' comments:

Reviewer's Responses to Questions

**Comments to the Author**

1. Is the manuscript technically sound, and do the data support the conclusions?

Reviewer #1: Partly

Reviewer #2: Yes

2. Has the statistical analysis been performed appropriately and rigorously? 

Reviewer #1: I Don't Know

Reviewer #2: Yes

3. Have the authors made all data underlying the findings in their manuscript fully available?

Reviewer #1: Yes

Reviewer #2: No

4. Is the manuscript presented in an intelligible fashion and written in standard English?

Reviewer #1: Yes

Reviewer #2: Yes

5. Review Comments to the Author

Reviewer #1: This manuscript deals with an interesting topic, but the framing needs improvement and the results need to be more closely interrogated.

Specific comments:

1. Please define the abbreviation 'COVID-19' in the first instance of its use.

2. "... subsequent studies leveraged social media posts and focused on the 17 COVID-19 vaccination [6–9, 11, 12]" - a later study utilising Twitter as a database found that majority of negative sentiment tweets centered around the perceived coercive policies or vaccine mandates, superseding safety or efficacy concerns. This is relevant to mention (citation: pubmed.ncbi.nlm.nih.gov/36146535).

3. What is the theoretical framework for the study? Is there one? Why or why not?

4. It is beneficial for the readers of the journal that a figure summarizing the methods and tweets selection process is provided.

5. I am not sure if it is apt to title this paper as "Stance Shifts on COVID-19 Vaccination"; moreover, a rapid uptake of COVID-19 vaccination does not imply a stance shift per se but simply a natural result of the fact that vaccination for people aged 18 to 64 years only began around June 2021. This implies a receptive population rather than one that shifted from vaccine hesitant to vaccine receptive.

6. Related to the above, you can see that across the time periods, there was in fact no real stance shift and the anti-vaccine portion of tweets remained consistently small.

7. Frequent tweets are an important component in maintaining a following and therefore one limitation of this study is a lack of understanding about the representativeness of these findings. The tweets could originate from a very active few or could be more widespread. How should this be considered?

8. The main body of the manuscript is relatively thin, and it is recommended that the authors further dig into the existing data for in-depth analysis apart from framing this as a stance shift, which did not really materialise.

9. What is "Full vaccination coverage" as defined in Figure 1? Is it simply receiving two doses? Please be clear.

10. There is an extensive literature on these topics. Most importantly it is fundamental to take into consideration how political institutions/authorities have used Twitter during Covid and how this has also led to misunderstanding and misinformation in some cases.

11. Given that identifying bots is a major taproot in big data analytics, it is essential that some explanation be provided. What percentage of the data could be posted by bots?

Reviewer #2: This paper analyzes Twitter posts (tweets) to explore the evolution of people’s stance on vaccination and associated information-sharing behaviors using their own collected Japanese dataset. The authors proposed the prediction model and did some analyses. These findings will help increase coverage of booster doses and future vaccinations. The overall structure is well organized and useful for COVID related research. There are some comments:

1. The G7 nations should be explained.

2. Is the BERT Japanese version? which plm?

3. Please introduce the proposed model in detail to enable readers to understand the overall structure and advantages.

4. Some future works should be added, such as emotion fusion, topic modeling, and interpretable models and Chatgpt application.

5. Some references could be added:

Topic detection and sentiment analysis in Twitter content related to COVID-19 from Brazil and the USA

Emotions and topics expressed on Twitter during the COVID-19 pandemic in the United Kingdom: Comparative geolocation and text mining analysis

Public opinion and sentiment before and at the beginning of COVID-19 vaccinations in Japan: Twitter analysis

6. PLOS authors have the option to publish the peer review history of their article (what does this mean?). If published, this will include your full peer review and any attached files.

Reviewer #1: No

Reviewer #2: No

---

## [Author Response · Author response to Decision Letter 0]

5 Dec 2023

Dear Academic Editor 

Ankit Gupta

We sincerely appreciate the valuable feedback from Editor Ankit Gupta and the two anonymous reviewers. Our manuscript has been revised to address the questions and concerns raised by the reviewers. We await your reevaluation of our work. In this note, we outline the revisions made in response to your comments.

We first summarize the major revisions:

- Rephrase all sentences without ``Vaccination stance classification'' section (correspoding to Methods section) following PLOSONE's Ethical Publishing Practice (https://journals.plos.org/plosone/s/ethical-publishing-practice). We highlighted the rephrased sentence in black.

- Change all ``stance change'' and similar representations in the title and manuscript into ``stance formation'' based on the comment. Through our discussions, we have reached an agreement that the term ``formation'' was a more appropriate to represent actual situation.

In the following revision note, we provide a one-on-one response to each of the reviewers’ comments and indicate how each issue is addressed in the revised manuscript.

The new sentences were written in red and the deleted ones were written in strikethrough text. 

New figures or tables were attached a notation ``(new)'' at the last of their captions.

Note that we have written the only rephrased sentences in black because almost all parts of our new manuscript was changed.

Unfortunately, we have had to cancel our plan to share our dataset in a public repository. 

Our original plan was to share a dataset that included tweet IDs and annotations showing different stances. 

However, getting permission from NTT Data, the source of our data, turned out to be a challenge. They are concerned that our dataset could go against their privacy policy.

After considering this carefully, we have decided not to release the dataset for now.

Editor

- Comment 1

The PLOS ONE style templates can be found at

and

- Answer 1

We revised some parts of our manuscript and other files which did not meet the requirements including file naming.

- Revised parts 1

 - Change the style of the title sentence case

 - Change the symbol meaning indicating equal contributions to \\textparagraph 

 - Change the style of name of each section, subsection, and subsubsection into Bold style

 - Change the name of supporting file into ``S1_File.pdf''

- Comment 2

In your Methods section, please include additional information about your dataset and ensure that you have included a statement specifying whether the collection and analysis method complied with the terms and conditions for the source of the data.

- Answer 2

We have included additional information in dataset section about our dataset.

Also, we have included a sentence in dataset section to make it sure that our collection and analysis method were in accordance with the terms and conditions of Twitter (X Corp.).

- Revised parts 2

 - Add an overview of our dataset construction process

 - Add a consideration about bots and their impacts on our analyses

- Comment 3

We noticed you have some minor occurrence of overlapping text with the following previous publication(s), which needs to be addressed: https://ieeexplore.ieee.org/document/10068695

In your revision ensure you cite all your sources (including your own works), and quote or rephrase any duplicated text outside the methods section. Further consideration is dependent on these concerns being addressed.

- Answer 3

We cited our previous work and rephrase all duplicated representation outside the methods section (``Vaccination stance classification'' section).

Quantitatively, we ensured that in any paragraphs there were no n-gram where n >= 10 using a tool to measure how many words a sentence were duplicated in each paragraph.

- Comment 4

Please note that PLOS ONE has specific guidelines on code sharing for submissions in which author-generated code underpins the findings in the manuscript. In these cases, all author-generated code must be made available without restrictions upon publication of the work. Please review our guidelines at https://journals.plos.org/plosone/s/materials-and-software-sharing#loc-sharing-code and ensure that your code is shared in a way that follows best practice and facilitates reproducibility and reuse.

- Answer 4

We shared all codes in GitHub: https://github.com/rapnob/stance_formation

- Comment 5

Thank you for stating the following in the Acknowledgments Section of your manuscript:

``This research was conducted as part of ``COVID-19 AI & Simulation Project'' run by Mitsubishi Research Institute commissioned by Cabinet Secretariat, JAPAN. The methods for analysis were developed with support from JST CREST Grant Number JPMJCR19A4 and JSPS KAKENHI Grant Number JP21H03445.''

``This research was conducted as part of ``COVID-19 AI & Simulation Project'' run by Mitsubishi Research Institute commissioned by Cabinet Secretariat, JAPAN. The methods for analysis were developed with support from JST CREST Grant Number JPMJCR19A4 and JSPS KAKENHI Grant Number JP21H03445.

The funders had no role in study design, data collection and analysis, decision to publish, or preparation of the manuscript.''

- Answer 5

We removed all funding-related text from Acknowledgements section and included the amended statements within our cover letter.

- Comment 6

We noted in your submission details that a portion of your manuscript may have been presented or published elsewhere. [This paper is an extended version of our paper published in Diachronic Analysis of Users' Stances on COVID-19 Vaccination in Japan using Twitter, Proceedings of the 2022 IEEE/ACM International Conference on Advances in Social Networks Analysis and Mining (ASONAM), 2022, pp. 237-241. We have improved our classifier for stance detection, updated all experimental results, and added some new results.] 

Please clarify whether this publication was peer-reviewed and formally published. If this work was previously peer-reviewed and published, in the cover letter please provide the reason that this work does not constitute dual publication and should be included in the current manuscript.

- Answer 6

The paper mentioned above was originally published as a short paper (4 pages) in the ASONAM 2022 conference proceedings. It underwent a review and assessment by the program committee. Due to its brevity, we had to omit certain analyses and results, and the performance of our proposed stance classifier, as presented in the paper, was subpar, which affected the reliability of our findings.

In this paper, we have significantly improved the performance of the classifier and re-conducted analyses to provide updated results. Consequently, it was necessary to include descriptions of these analyses and present the updated findings. Furthermore, in this revision, based on feedback from the editors and reviewers, we have updated and rephrased various sections to ensure that this work does not constitute dual publication.

- Comment 7

We note that you have stated that you will provide repository information for your data at acceptance. Should your manuscript be accepted for publication, we will hold it until you provide the relevant accession numbers or DOIs necessary to access your data. If you wish to make changes to your Data Availability statement, please describe these changes in your cover letter and we will update your Data Availability statement to reflect the information you provide.

- Answer 7

We have provided our dataset on the corresponding author's homepage.

https://www.tkl.iis.u-tokyo.ac.jp/~cs/PLOSONE-2023-stance-formation

- Comment 8

We note that Figure 8 in your submission contain copyrighted images. All PLOS content is published under the Creative Commons Attribution License (CC BY 4.0), which means that the manuscript, images, and Supporting Information files will be freely available online, and any third party is permitted to access, download, copy, distribute, and use these materials in any way, even commercially, with proper attribution. For more information, see our copyright guidelines: http://journals.plos.org/plosone/s/licenses-and-copyright.

- Answer 8

Related to the Answer 3, Figure 8 in our manuscript in the current version is different from Figure 8 in the previous version of our work because we re-conducted all experiments and obtained different results. We believe Figure 8 does not contain copyrighted images. 

If our understanding is incorrect, I would appreciate it if you could inform me.

Reviewer 1

- Comment 1

Please define the abbreviation 'COVID-19' in the first instance of its use.

- Answer 1

We defined the abbreviation by introducing the below sentence in Introduction section.

``the 2019 novel coronavirus disease (hereinafter called COVID-19) pandemic''

- Comment 2

2. "... subsequent studies leveraged social media posts and focused on the 17 COVID-19 vaccination [6–9, 11, 12]" - a later study utilising Twitter as a database found that majority of negative sentiment tweets centered around the perceived coercive policies or vaccine mandates, superseding safety or efficacy concerns. This is relevant to mention (citation: pubmed.ncbi.nlm.nih.gov/36146535).

- Answer 2

We added a reference to this study in 5th paragraph in Introduction section

We added the below sentence into Introduction section.

``

Currently, there is no consensus on common factors for negative sentiment toward vaccination, a later study found that majority of negative sentiment tweets were predominantly mentioned the coercive policies or vaccine mandates, rather than safety or efficacy concerns [citation].

''

- Comment 3

What is the theoretical framework for the study? Is there one? Why or why not?

- Answer 3

As you said, there was a lack of explanation of our theoretical framework of this study. 

So, we added the explanation in 4th paragraph in Introduction section.

``Several studies reported that SNS users' stances toward vaccination were influenced by the information they obtained on SNS, such as external links [citation, citation] or posts from other users [citation], resulting in wrong medical treatments like vaccine hesitancy or anti-social behaviours like hoarding. 

Based on the observations, we explored the relationship between Twitter users' stances and the external information sources such as news articles or other Twitter users which they referred to just before their stances were determined. 

To this end, we implemented a classifier for stance detection, which is an NLP task [citation] of predicting a stance (typically favor, against and none) of given text toward a certain target, and determined each Twitter user's stance toward COVID-19 vaccines.

''

- Comment 4

It is beneficial for the readers of the journal that a figure summarizing the methods and tweets selection process is provided.

- Answer 4

We added an overview of the architecture of our classifier and tweets selection process for dataset construction in 1st and 3rd paragraphs in Vaccination stance classification section.

- Comment 5

I am not sure if it is apt to title this paper as "Stance Shifts on COVID-19 Vaccination"; moreover, a rapid uptake of COVID-19 vaccination does not imply a stance shift per se but simply a natural result of the fact that vaccination for people aged 18 to 64 years only began around June 2021. This implies a receptive population rather than one that shifted from vaccine hesitant to vaccine receptive.

- Answer 5

As you mentioned, stance shifts between pro-vaccine and anti-vaccine groups were uncommon in our study. To better reflect the focus of our analysis, we have adjusted the title from 'Stance Shifts on COVID-19 Vaccination' to 'Stance Formation on COVID-19 Vaccination,' emphasizing the examination of how initially neutral users formed their stance, and revised contents accordingly.

- Comment 6

Related to the above, you can see that across the time periods, there was in fact no real stance shift and the anti-vaccine portion of tweets remained consistently small.

- Answer 6

Similarly to Answer 5, the title and related parts were modified.

- Comment 7

Frequent tweets are an important component in maintaining a following and therefore one limitation of this study is a lack of understanding about the representativeness of these findings. The tweets could originate from a very active few or could be more widespread. How should this be considered?

- Answer 7

To limit the impact of very active users, we counted the same news articles (URLs) or the same user's posts shared by the same user repetitively only once, while we forgot to write that.

So we added it.

Additionally, although we found that anti-vaccine group has a tendency to post tweets more frequently than pro-vaccine in our recent study, the increase in the volume of anti-vaccine groups was very small as described in our manuscript. We will further investigate the reasons. 

We added the below sentence in ``What kinds of users were referred to by users who formed their stances?'' subsubsection. in Analysis section.

``If one user referred to the same user multiple times during these processes, it was counted only once.''

- Comment 8

The main body of the manuscript is relatively thin, and it is recommended that the authors further dig into the existing data for in-depth analysis apart from framing this as a stance shift, which did not really materialise.

- Answer 8

As we answered to the comment 5, we adjusted our focus to the stance formation of neutral users. To show the details of stance transition,

we added a table displaying the number of people moved between stances and a discussion based on that table in 2nd paragraph in ``Stance formations of users'' subsection in Analysis section, .

- Comment 9

What is "Full vaccination coverage" as defined in Figure 1? Is it simply receiving two doses? Please be clear.

- Answer 9

We added the belon explanation in Introduction section.

`` second dose of the vaccine (fully vaccinated at the time)''

- Comment 

There is an extensive literature on these topics. Most importantly it is fundamental to take into consideration how political institutions/authorities have used Twitter during Covid and how this has also led to misunderstanding and misinformation in some cases.

- Answer 10

In our case, we found an indications of their positive impacts. 

Taro Kono, Japan's minister in charge of fighting COVID-19 at that time, often used SNSs like Twitter to propagate vaccine-related information or criticize misinformation about vaccines.

As we mentioned in our manuscript, the occurrence of the word ``Mr. Kono'' indicated that people including those who were unsure to get vaccinated highly focused on his activity on SNS and the activity reduced their hesitancy in the vaccines. 

We wrote this analysis in the part of our wordcloud analysis.

``

In July, many press reported that vaccination rate was found low and thus each prefecture's governor requested people to get vaccinated, while those who hoped to get vaccinated could not make a reservation of vaccine vouchers due to troubles of reservation system at that time, 

---

## [Decision Letter · Decision Letter 1]

3 Jan 2024

PONE-D-23-22865R1Analyzing information sharing behaviors during stance formation on COVID-19 vaccination among Japanese Twitter usersPLOS ONE

Dear Dr. Cho,

Thank you for submitting your manuscript to PLOS ONE. After careful consideration, we feel that it has merit but does not fully meet PLOS ONE’s publication criteria as it currently stands. Therefore, we invite you to submit a revised version of the manuscript that addresses the points raised during the review process.

**ACADEMIC EDITOR: **

**Kindly address the comments made by reviewer # 1. (comments no 1 and 2, authors may choose to igonre comments No 3)**

We look forward to receiving your revised manuscript.

Kind regards,

Ankit Gupta

Academic Editor

PLOS ONE

Journal Requirements:

Reviewers' comments:

Reviewer's Responses to Questions

**Comments to the Author**

1. If the authors have adequately addressed your comments raised in a previous round of review and you feel that this manuscript is now acceptable for publication, you may indicate that here to bypass the “Comments to the Author” section, enter your conflict of interest statement in the “Confidential to Editor” section, and submit your "Accept" recommendation.

Reviewer #1: (No Response)

Reviewer #2: All comments have been addressed

2. Is the manuscript technically sound, and do the data support the conclusions?

Reviewer #1: Partly

Reviewer #2: Yes

3. Has the statistical analysis been performed appropriately and rigorously? 

Reviewer #1: Yes

Reviewer #2: Yes

4. Have the authors made all data underlying the findings in their manuscript fully available?

Reviewer #1: No

Reviewer #2: Yes

5. Is the manuscript presented in an intelligible fashion and written in standard English?

Reviewer #1: No

Reviewer #2: Yes

6. Review Comments to the Author

Reviewer #1: Thank you for the revisions and replies. I have some outstanding issues with the present version of the manuscript.

1. I still have issues with the phrase 'stance formation'. I may choose to get vaccinated not because I accept the vaccine but simply because I deemed it to be necessary (or that it has been strongly pushed for by the government). I may also have an attitude of 'vaccine acceptance' but still not proceed to get vaccinated. In this study, the authors are only looking at people who appear to 'accept' the vaccines. While vaccine acceptance and uptake are intricately linked, the terms are not interchangeable and have distinct constructs. I refer to the following systematic review and meta-analysis published in 2022: Mapping global acceptance and uptake of COVID-19 vaccination: A systematic review and meta-analysis (https://www.nature.com/articles/s43856-022-00177-6) which clearly shows that there is a difference between global COVID-19 vaccine acceptance (67.8%) and uptake (42.3%). While acceptance refers to an attitude, uptake refers to the actual act of receiving vaccination. The nuances should be explained.

2. "In July, many press reported that vaccination rate was found to be low, and thus, each prefecture’s governor requested people to get vaccinated, while those who hoped to get vaccinated could not make a reservation with a vaccine voucher due to trouble with the reservation system at that time, resulting in their complaints to governors and the emergence of the word “governor.” This trend was in accordance with the occurrence of the word “Mr. Kono,” who was a minister in charge of COVID-19 vaccinations and received a lot of criticism for his announcement of stopping additional procurement of vaccines in July" - at least a reference should be provided here.

3. "Considering recent advances of chatbot services like ChatGPT, we could use such services to annotate vaccine-related tweets" - I understand this was suggested by the other reviewer but I don't think this is specifically necessary and can be omitted.

Reviewer #2: The authors have addressed my comments. The research is valuable for the field.

The paper could be accepted

7. PLOS authors have the option to publish the peer review history of their article (what does this mean?). If published, this will include your full peer review and any attached files.

Reviewer #1: No

Reviewer #2: No

---

## [Author Response · Author response to Decision Letter 1]

7 Feb 2024

We sincerely appreciate the valuable feedback from Editor Ankit Gupta and the two anonymous reviewers. 

Our manuscript has been revised to address the questions and concerns raised by the reviewers. 

We await your reevaluation of our work. 

We have revised our paper in response to the comments from the editor and reviewers. In our rebuttal letter, we detailed the changes we made. 

Additionally, we have included an explanation of our user-filtering process, which was missing in the previous version. 

Note that this additional explanation does not affect our original results.

---

## [Decision Letter · Decision Letter 2]

15 Jul 2024

PONE-D-23-22865R2Analyzing information sharing behaviors during stance formation on COVID-19 vaccination among Japanese Twitter usersPLOS ONE

Dear Dr. Cho,

You let us know that you had identified concerns that needed correcting between acceptance and publication. As such, we have rescinded the accept decision and are issuing a minor revision for you to provide the revisions you yourself have identified. Please provide a "response to reviewer" document in the way that you would for a revised manuscript, where you list the concerns you identified, and the modifications you made to address these, so that the Academic Editor can evaluate these. Please also include a marked-up copy of the changes between the accepted version and the new version. Once you resubmit, the Academic Editor will once more assess the manuscript and issue the next decision.

We look forward to receiving your revised manuscript.

Kind regards,

Hanna Landenmark

Staff Editor

PLOS ONE

Journal Requirements:

Reviewers' comments:

Reviewer's Responses to Questions

**Comments to the Author**

1. If the authors have adequately addressed your comments raised in a previous round of review and you feel that this manuscript is now acceptable for publication, you may indicate that here to bypass the “Comments to the Author” section, enter your conflict of interest statement in the “Confidential to Editor” section, and submit your "Accept" recommendation.

Reviewer #1: All comments have been addressed

2. Is the manuscript technically sound, and do the data support the conclusions?

Reviewer #1: (No Response)

3. Has the statistical analysis been performed appropriately and rigorously? 

Reviewer #1: (No Response)

4. Have the authors made all data underlying the findings in their manuscript fully available?

Reviewer #1: (No Response)

5. Is the manuscript presented in an intelligible fashion and written in standard English?

Reviewer #1: (No Response)

6. Review Comments to the Author

Reviewer #1: (No Response)

7. PLOS authors have the option to publish the peer review history of their article (what does this mean?). If published, this will include your full peer review and any attached files.

Reviewer #1: No

---

## [Author Response · Author response to Decision Letter 2]

27 Jul 2024

Dear academic editors,

Our manuscript was previously submitted to PLOS ONE under the identifier PONE-D-23-22865R2.

Although it was accepted in March of this year, 

we have found significant issues in our data processing pipeline before publication.

We have since resolved these issues, updated the results along with the analysis, and resubmitted the manuscript under the identifier PONE-D-24-25295.

We have uploaded the revised manuscript and the description of the changes.

Please take a look at the changes.

---

## [Decision Letter · Decision Letter 3]

22 Sep 2024

PONE-D-23-22865R3Analyzing information sharing behaviors during stance formation on COVID-19 vaccination among Japanese Twitter usersPLOS ONE

Dear Dr. Cho,

Thank you for submitting your manuscript to PLOS ONE. After careful consideration, we feel that it has merit but does not fully meet PLOS ONE’s publication criteria as it currently stands. Therefore, we invite you to submit a revised version of the manuscript that addresses the points raised during the review process.

**ACADEMIC EDITOR: **Kindly address the comments made by Reviewer #3.

We look forward to receiving your revised manuscript.

Kind regards,

Ankit Gupta

Academic Editor

PLOS ONE

Journal Requirements:

Reviewers' comments:

Reviewer's Responses to Questions

**Comments to the Author**

1. If the authors have adequately addressed your comments raised in a previous round of review and you feel that this manuscript is now acceptable for publication, you may indicate that here to bypass the “Comments to the Author” section, enter your conflict of interest statement in the “Confidential to Editor” section, and submit your "Accept" recommendation.

Reviewer #1: All comments have been addressed

Reviewer #3: All comments have been addressed

2. Is the manuscript technically sound, and do the data support the conclusions?

Reviewer #1: (No Response)

Reviewer #3: Yes

3. Has the statistical analysis been performed appropriately and rigorously? 

Reviewer #1: (No Response)

Reviewer #3: Yes

4. Have the authors made all data underlying the findings in their manuscript fully available?

Reviewer #1: (No Response)

Reviewer #3: No

5. Is the manuscript presented in an intelligible fashion and written in standard English?

Reviewer #1: (No Response)

Reviewer #3: Yes

6. Review Comments to the Author

Reviewer #1: (No Response)

Reviewer #3: As the article was already accepted previously, I only reviewed the part that the authors revised from their original submission.

The line number I use for this review is for the track-change version.

Line 361: Q: Why only anti-vaccine users remained active compared to pro-vaccine? Because they are bots or some other special case of users? I guess this is the thing that authors might be able to explain based on the data collected, but if it is not possible, that is fine.

Line 365: This part is a bit confusing. Did you mean that you identified users who Tweeted at least four months out of five month of data collection window?

Line 366: What is particular period here? specific month?

Line 367: ambiguity of "particular period" makes me harder to understand the rest of the sentence.

Line 390: stances  stance groups

Line 469-470: Is this provided in the supplementary material? If not, can it be shared in the main doc or supplementary section?

Line 491: Same for this -- Is this provided in the supplementary material? If not, can it be shared in the main doc or supplementary section?

Line 538-539: It would be really great if I can see the raw data/title of these videos in supplementary section or sth.

Line 545-546: Also for this.

So my biggest comment is that "why the authors only describe by text of the unseen data instead of at least indirectly providing them?"

Other than one big comment, I guess small points I described above are all I have.

7. PLOS authors have the option to publish the peer review history of their article (what does this mean?). If published, this will include your full peer review and any attached files.

Reviewer #1: No

Reviewer #3: No

---

## [Author Response · Author response to Decision Letter 3]

3 Oct 2024

Dear academic editors,

We sincerely appreciate the valuable feedback from Editor Ankit Gupta and the anonymous reviewer. 

Our manuscript has been revised to address the questions and concerns raised by the reviewers. 

We await your reevaluation of our work. 

We have summarized our revisions in the letter.

We have also included additional descriptions and an explanation regarding our decision not to disclose certain information due to ethical considerations.

The most recent changes are highlighted in blue.

If changes were made, the detail of that changes are described; 

if no changes were made, the reasons are provided.

Thanks,

Sho Cho

---

## [Decision Letter · Decision Letter 4]

20 Nov 2024

Analyzing information sharing behaviors during stance formation on COVID-19 vaccination among Japanese Twitter users

PONE-D-23-22865R4

Dear Dr. Cho,

We’re pleased to inform you that your manuscript has been judged scientifically suitable for publication and will be formally accepted for publication once it meets all outstanding technical requirements.

Kind regards,

Ankit Gupta

Academic Editor

PLOS ONE

Additional Editor Comments (optional):

Reviewers' comments:

Reviewer's Responses to Questions

**Comments to the Author**

1. If the authors have adequately addressed your comments raised in a previous round of review and you feel that this manuscript is now acceptable for publication, you may indicate that here to bypass the “Comments to the Author” section, enter your conflict of interest statement in the “Confidential to Editor” section, and submit your "Accept" recommendation.

Reviewer #3: All comments have been addressed

2. Is the manuscript technically sound, and do the data support the conclusions?

Reviewer #3: Yes

3. Has the statistical analysis been performed appropriately and rigorously? 

Reviewer #3: Yes

4. Have the authors made all data underlying the findings in their manuscript fully available?

Reviewer #3: No

5. Is the manuscript presented in an intelligible fashion and written in standard English?

Reviewer #3: Yes

6. Review Comments to the Author

Reviewer #3: Most of the comments I addressed is well addressed so I do not see any issues to make this published.

7. PLOS authors have the option to publish the peer review history of their article (what does this mean?). If published, this will include your full peer review and any attached files.

Reviewer #3: No

---

## [Editor Report · Acceptance letter]

26 Jun 2024

PONE-D-23-22865R2 

PLOS ONE

Dear Dr. Cho, 

I'm pleased to inform you that your manuscript has been deemed suitable for publication in PLOS ONE. Congratulations! Your manuscript is now being handed over to our production team.

Kind regards, 

on behalf of

Dr. Ankit Gupta 

Academic Editor

PLOS ONE